# Selective sorting of microRNAs into exosomes by phase-separated YBX1 condensates

**Xiao-Man Liu, Liang Ma, Randy Schekman***

Department of Molecular and Cell Biology, Howard Hughes Medical Institute, University of California, Berkeley, Berkeley, United States

**Abstract** Exosomes may mediate cell-to-cell communication by transporting various proteins and nucleic acids to neighboring cells. Some protein and RNA cargoes are significantly enriched in exosomes. How cells efficiently and selectively sort them into exosomes remains incompletely explored. Previously, we reported that YBX1 is required in sorting of miR-223 into exosomes. Here, we show that YBX1 undergoes liquid-liquid phase separation (LLPS) *in vitro* and in cells. YBX1 condensates selectively recruit miR-223 *in vitro* and into exosomes secreted by cultured cells. Point mutations that inhibit YBX1 phase separation impair the incorporation of YBX1 protein into biomolecular condensates formed in cells, and perturb miR-233 sorting into exosomes. We propose that phase separation-mediated local enrichment of cytosolic RNA-binding proteins and their cognate RNAs enables their targeting and packaging by vesicles that bud into multivesicular bodies. This provides a possible mechanism for efficient and selective engulfment of cytosolic proteins and RNAs into intraluminal vesicles which are then secreted as exosomes from cells.

## Editor's evaluation

This paper represents a significant step forward in our understanding how proteins and RNAs are selectively loaded into exosomes, and describes an unexpected cellular use of biomolecular phase separation. This paper will be of interest to molecular cell biologists who study extracellular vesicle biology and liquid-liquid phase separation.

*For correspondence:
schekman@berkeley.edu

## Introduction

Extracellular vesicles (EVs) secreted into the extracellular space appear to mediate some forms of intercellular communication (*Colombo et al., 2014*; *Maia et al., 2018*; *Song et al., 2021*). Different sub-populations of EVs bud from the plasma membrane or arise from membrane internalized into endosomes to form multi-vesicular bodies (MVB) that fuse at the cell surface to secrete intralumenal vesicles (ILV). Secreted ILVs, referred to as exosomes, are typically 30–150 nm vesicles with a buoyant density of ~1.10–1.19 g/ml (*Mincheva-Nilsson et al., 2016*). Plasma membrane-derived microvesicles, also referred to as shedding vesicles, are more heterogeneous with sizes ranging from 30 to 1000 nm (*Cocucci et al., 2009*; *Raposo and Stoorvogel, 2013*). During their biogenesis, EVs may selectively capture proteins, lipids, metabolites, and nucleic acids which vary according to the cell of origin.

The selectivity for cargo sorting into EVs is best studied for RNA molecules. Several RNA-binding proteins (RBPs), including heterogeneous nuclear ribonucleoproteins A2/B1 (hnRNPA2B1) (*Villarroya-Beltri et al., 2013*), SYNCRIP (*Hobor et al., 2018*; *Santangelo et al., 2016*), HuR (*Mukherjee et al., 2016*) and major vault protein (MVP) (*Statello et al., 2018*; *Teng et al., 2017*), have been implicated

in the sorting of RNAs into EVs. In these studies, extracellular vesicles were isolated by sedimentation at ~100,000 xg. These crude EV preparations contain heterogeneous populations of vesicles and membrane-free ribonucleoprotein particles (RNPs), which has complicated the study of requirements for sorting selectivity. To solve this problem, our lab developed a buoyant density-based procedure to resolve EVs into two fractions and found that certain miRNAs are highly enriched in exosomes as opposed to EVs of lower buoyant density (*Shurtleff et al., 2016*; *Temoche-Diaz et al., 2020*). We further demonstrated that an RBP, YBX1, is required for selective sorting of miR-223 into exosomes in a cell-free reaction that recapitulates miRNA sorting into vesicles and in cultured cells (*Shurtleff et al., 2016*). In subsequent work, we identified another RBP, Lupus La protein, that is required for selective sorting of miR-122 into exosomes in the breast cancer cell line, MDA-MB-231 (*Temoche-Diaz et al., 2019*). However, the means by which certain RNPs are efficiently and selectively packaging into exosomes remains unclear.

Eukaryotic cells form compartments that contain both membrane-bound organelles and non-membrane-bound organelles to optimize the efficiency of biological processes (*Wheeler and Hyman, 2018*). Membraneless organelles, also referred to as biomolecular condensates, are assembled via liquid-liquid phase separation (LLPS), a process in which molecules such as proteins, RNA and other biopolymers are concentrated into a liquid-like compartment. Examples include cytoplasmic condensates such as stress granules, processing bodies (P-bodies), germline P-granules and nuclear condensates such as the nucleolus, Cajal bodies, and paraspeckles. The constituents in these condensates exhibit high mobility and rapidly exchange with the surrounding cytoplasm or nucleoplasm (*Banani et al., 2017*; *Shin and Brangwynne, 2017*; *Zhao and Zhang, 2020*). Cells also contain more stable gel-liked condensates, such as Balbiani bodies, centrosomes, nuclear pores, and amyloid bodies. These condensates exhibit slower rearrangement and fusion (*Spannl et al., 2019*; *Woodruff et al., 2018*). LLPS is mediated by weak, transient interactions conferred by proteins with intrinsically disordered regions (IDRs) and/or multivalent domains (*Molliex et al., 2015*; *Shin and Brangwynne, 2017*; *Wheeler and Hyman, 2018*). IDRs lack a fixed or ordered three-dimensional structure and often comprise biased amino acids, in particular polar and charged residues, including glycine, serine, glutamine, arginine and lysine, and aromatic residues (e.g. tyrosine and phenylalanine) (*Shin and Brangwynne, 2017*). Amino acid side chain charge-charge, charge-π, and π-π stacking interactions have been implicated in LLPS condensate formation (*Brangwynne et al., 2015*). Protein-RNA and RNA-RNA interactions also contribute to RNP condensates such as stress granules and P-bodies (*Tauber et al., 2020*; *Yang et al., 2020*). Liquid-like condensates may harden over time into less fluid structures, such as hydrogels. Aberrant phase transition may be the basis of certain neurodegenerative disorders and cancer (*Alberti and Hyman, 2016*; *Nedelsky and Taylor, 2019*; *Taylor et al., 2016*).

Phase-separated condensates exhibit selective properties, favoring some proteins and RNAs and excluding others (*Alberti et al., 2019*). P-bodies, for example, are cytoplasmic RNA granules formed by condensation of translationally repressed mRNAs associated with proteins related to mRNA decay (*Luo et al., 2018*; *Parker and Sheth, 2007*; *Teixeira et al., 2005*). P-body proteomes are enriched in proteins containing IDRs (*Youn et al., 2019*), including those involved in mRNA decapping and decay and miRNA/siRNA silencing (*Luo et al., 2018*). MiRNAs, their cognate mRNAs and proteins related to miRNA-mediated suppression, including Ago proteins, GW182, Rck, and MOV10 are concentrated in P-bodies (*Eystathioy et al., 2003*; *Kulkarni et al., 2010*; *Liu et al., 2005a*; *Liu et al., 2005b*; *Sen and Blau, 2005*).

Phase-separated condensates have been implicated in the membrane enclosure of cytosolic proteins associated with autophagy. Cargo proteins such as the Ape1 complex, the PGL granule, and p62, the autophagy cargo receptor, form phase-separated protein condensates, triggering formation of surrounding autophagosomes to ensure their specific and efficient transport to lysosomes or the vacuole (*Sun et al., 2018*; *Yamasaki et al., 2020*; *Zhang et al., 2018*). By extension, it seems likely that RNA-binding proteins may form condensates as a precursor to engulfment by endosomal membranes. Indeed, YBX1 contains an IDR sequence and was reported to be associated with P-bodies (*Yang and Bloch, 2007*) and involved in stress granule formation (*Lyons et al., 2016*; *Somasekharan et al., 2015*).

In this study, we report evidence that the YBX1 protein efficiently forms liquid-like droplets *in vitro* and in cells. We observed that miR-223 but not miR-190 or miR-144 efficiently partitioned into YBX1 droplets. YBX1 condensate formation required the C-terminal intrinsically disordered region

(IDR), dependent on the aromatic residue tyrosine and positive charged residues arginine and lysine. Point mutations that rendered YBX1 unable to phase separate disrupted YBX1 condensates in cells, interfered with recruitment of miR-223 to YBX1 droplets *in vitro*, and resulted in a failure of packaging of miR-223 into exosomes secreted by cells. We found that YBX1 condensed into processing bodies that contain several other proteins that were sorted into exosomes. We suggest that YBX1 liquid-like condensates may increase the local concentration of YBX1 molecules and bound RNA, and thereby direct the selective sorting miRNAs into exosomes, thus coupling RNP granules to RNA packaging into exosomes.

## Results

### YBX1 forms liquid-like condensates in cells

We have previously shown that an RBP, YBX1, is present in purified exosomes and is required for sorting miRNAs into exosomes (*Shurtleff et al., 2016*). To further characterize how YBX1 functions in this process, we first examined its subcellular localization. Endogenous YBX1 was observed concentrated in puncta in the cytosol, as visualized by the use of specific YBX1 antibody and immunofluorescence (IF) in fixed cells (*Figure 1A*). We hypothesized that the YBX1 puncta are liquid-like condensates that rapidly exchange their constituent molecules with the surrounding cytosol. To investigate this possibility, we constructed a stable cell line with YFP-tagged YBX1 that showed similar puncta as endogenous YBX1. Using this cell line, we performed fluorescence recovery after photobleaching (FRAP) experiments and found that after photobleaching, approximately 70 % of fluorescence was recovered within 90 s, suggesting a liquid-like behavior (*Figure 1B*). A compound previously shown to disrupt liquid-liquid phase separation of biomolecules (1,6-hexanediol, *Kroschwald et al., 2017*) caused YBX1 puncta to disassemble in a time- and concentration-dependent manner (*Figure 1C, D and E*). Another aliphatic alcohol, 2,5-hexanediol, which was reported to be much less active in the dissolution of FUS hydrogel droplets *in vitro* (*Lin et al., 2016*), was similarly less efficient in dissolving YBX1 puncta in cells (*Figure 1D and E*). We further observed two YBX1 puncta coalesced to form a larger punctum within a few seconds, reflecting the liquid-like property of puncta (*Figure 1F*). Thus, these data suggest that YBX1 forms a liquid-like biomolecular condensate in cells.

### YBX1 undergoes liquid-liquid phase separation (LLPS) *in vitro*

To address whether YBX1 exhibits liquid-like properties *in vitro*, we purified full-length recombinant human YBX1 protein fused to mGFP expressed in insect cells using the FlexiBAC system (*Lemaitre et al., 2019*) and performed phase separation assays with or without a molecular crowding agent, dextran (*Figure 2A*). We detected phase separation of YBX1 protein at increasing protein concentrations (*Figure 2A*). YBX1 started to form weak condensates at 3 µM and liquid droplets at 5 µM which corresponded to its approximate physiological cytosolic concentration in cells (4.6 µM) (*Itzhak et al., 2016*). We also observed that YBX1 droplets were rapidly dispersed by 1,6-hexanediol treatment (*Figure 2B*). As we observed in cells, two pure YBX1 droplets coalesced to form a larger spherical droplet, reflecting the nature of liquid-like droplets to minimize surface area by decreasing the surface/vol (*Figure 2C*). Correspondingly, in FRAP analysis, the mGFP::YBX1 fluorescence signal almost completely recovered 9 s after bleaching, consistent with the fluid state of YBX1 in granules in cells (*Figure 2D and E*). These data suggest that GFP-YBX1 phase separates to form liquid condensates *in vitro*. This behavior is intrinsic to YBX1 as the GFP molecule does not phase separate at these concentrations (*Kanaan et al., 2020*; *Pak et al., 2016*).

### YBX1 LLPS is likely driven by tyrosine-arginine residues in intrinsically disordered region (IDR)

YBX1 has three major domains: An N-terminal alanine/proline-rich (A/P) domain, a central cold shock domain (CSD), and a C-terminal domain (CTD) (*Figure 3A*). YBX1 is predicted to contain IDRs in both the N-terminal and C-terminal domains (*Figure 3A* and *Figure 3—figure supplement 1*). The long, C-terminal segment contains positively and negatively charged clusters of amino acids (*Figure 3A*). To identify the role of individual domains of YBX1 to form puncta in cells, we generated a series of constructs in which the A/P domain, CSD or the CTD were either deleted or expressed exclusively (*Figure 3B* and *Figure 3—figure supplement 2A*). Deletion of the CTD (YBX1-Δ128–324) completely

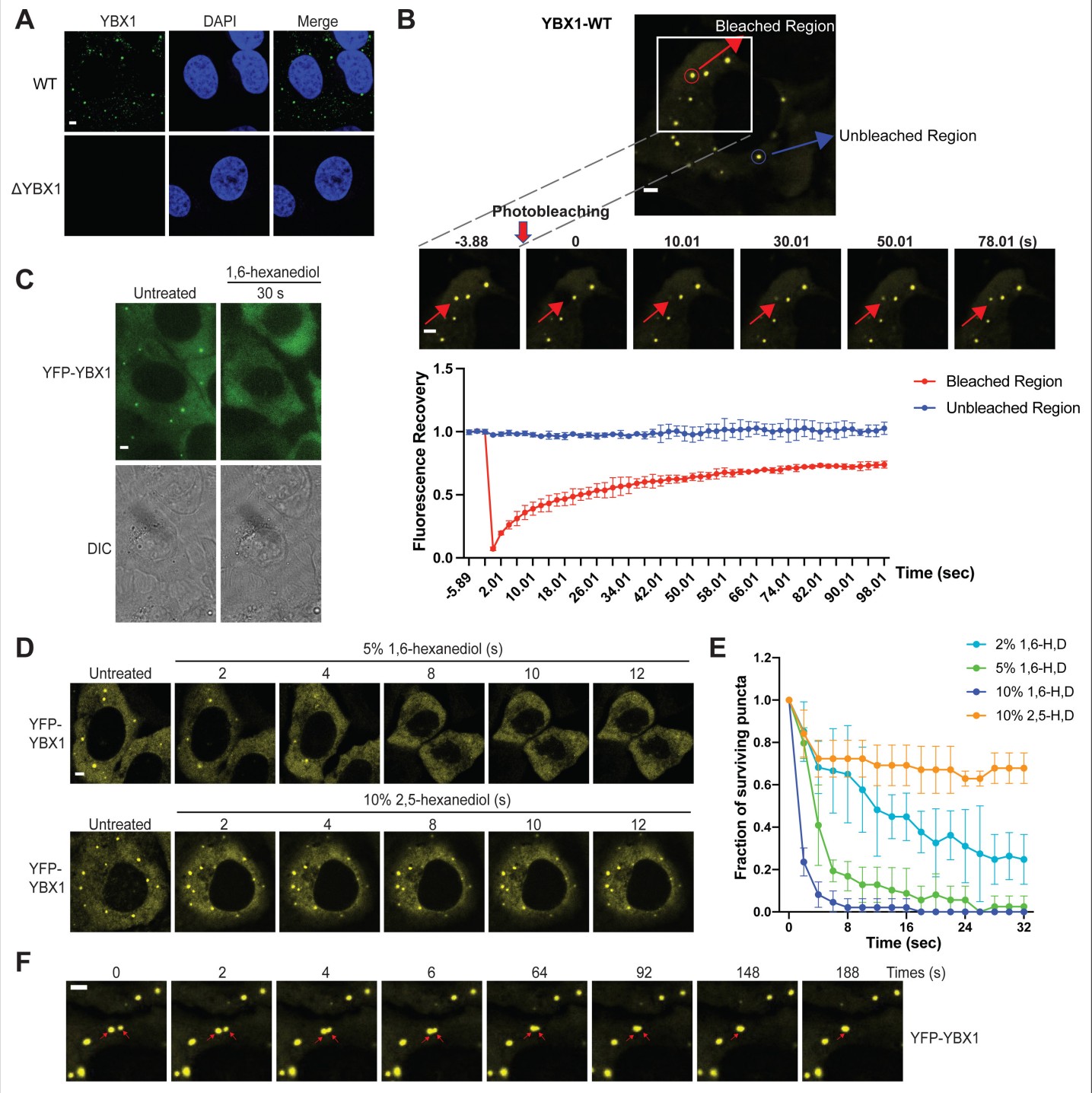

**Figure 1.** YBX1 forms liquid-like condensates in cells. (**A**) Subcellular localization of YBX1 in WT and ΔYBX1 from U2OS cells as visualized by YBX1 antibody. DAPI staining (blue) indicates the location of nuclei. (**B**) FRAP images show recovery of YFP-YBX1 puncta after photobleaching. U2OS cells with stable expression of YFP-YBX1 was subjected to FRAP analysis. The inset images (middle) are the representative FRAP images. The recovery kinetics of YFP-YBX1 are shown in the bottom. Error bars represent standard errors with n = 3. (**C**) The effect of 10 % 1,6-hexanediol on YFP-YBX1 puncta in U2OS cells. This image was performed on ECLIPSE TE2000 microscope at room temperature. (**D**) Fluorescence images of YFP-YBX1 in U2OS cells after treatment with 5 % 1,6-hexanediol or 10 % 2,5- hexanediol. Live cell imaging was performed on an LSM880 microscope with the incubation chamber maintained at 37 °C and 5 % $CO_2$. (See Materials and methods in details). (**E**) Number of YFP-YBX1 puncta in U2OS cells surviving over time after treatment with 1,6-hexanediol and 2,5-hexanediol. Error bars represent standard errors with n = 3. (**F**) Representative images of YBX1 puncta coalescence from U2OS cells. This live cell imaging was performed on an LSM880 microscope with the incubation chamber maintained 37 °C and 5 % $CO_2$. Scale bars, 3 μm.

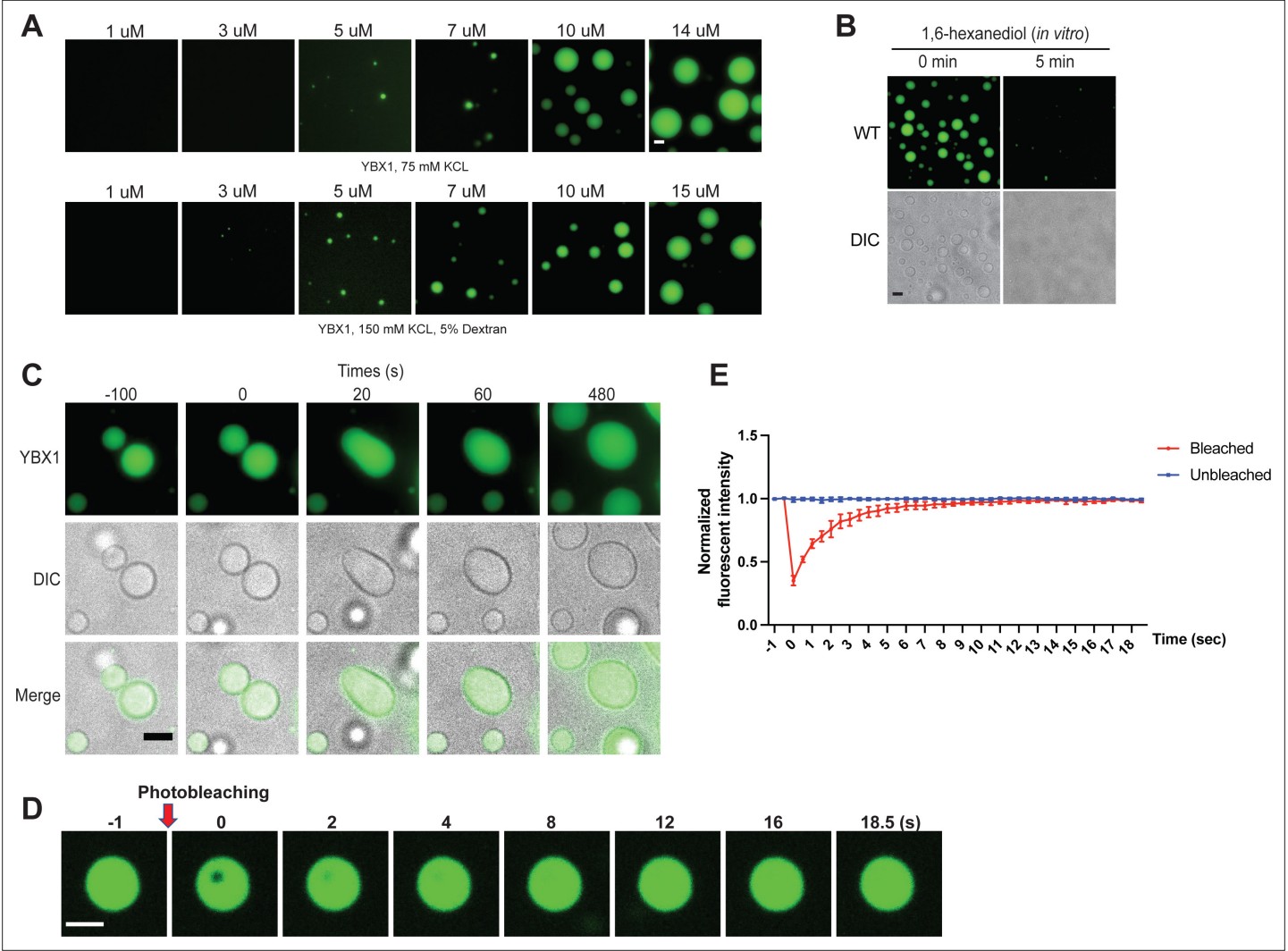

**Figure 2.** YBX1 forms liquid-like droplets *in vitro*. (**A**) Phase separation of YBX1 at different concentrations with or without addition of a crowding agent. Phase separation was induced by diluting the salt concentration from 500 mM to 75 mM or 150 mM in this assay. (**B**) The effect of 10 % 1,6-hexanediol on YBX1 droplets *in vitro*. Phase separation was induced by diluting the salt concentration from 500 mM to 75 mM in this assay. (**C**) Representative images of YBX1 droplets coalescence *in vitro*. Phase separation was induced by diluting the salt concentration from 500 mM to 75 mM in this assay. (**D**, **E**) Images (**D**) and quantification (**E**) of recovery of YBX1 droplets after photobleaching. A representative result of three independent experiments is shown. Phase separation was induced by diluting the salt concentration from 500 mM to 75 mM in this assay. Error bars represent standard errors. Scale bars, 3 μm.

blocked the formation of YBX1 puncta whereas the N-terminal A/P domain (YBX1-Δ1–55) was dispensable for puncta formation (*Figure 3C* and *Figure 3—figure supplement 2A and B*), suggesting that the C-terminal IDR rather than the N-terminal IDR was required for YBX1 condensate formation. Interestingly, the CTD (YBX1-Δ1–127) expressed alone was predominantly localized to the nucleus, likely within the nucleolus (*Figure 3—figure supplement 2A and B*), possibly due to nuclear export signals (NES) within the YBX1 N-terminal domain (*van Roeyen et al., 2013*). The CSD (YBX1-CSD) showed an even cellular distribution and a deletion lacking CSD (YBX1-ΔCSD) was predominantly nuclear (*Figure 3—figure supplement 2A and B*). We next studied the effect of these mutations on YBX1 droplet formation *in vitro*. Consistent with our observation of transfected cells, we found that CTD was indispensable for YBX1 LLPS. YBX1-CTD alone (YBX1-Δ1–127) also formed liquid-like droplets, suggesting that the C-terminal IDR was sufficient and essential for YBX1 LLPS (*Figure 3D* and *Figure 3—figure supplement 2E*).

Specific residues within IDRs have previously been reported to be involved in condensate formation (*Shin and Brangwynne, 2017*), thus we analyzed the residue distribution within this domain of

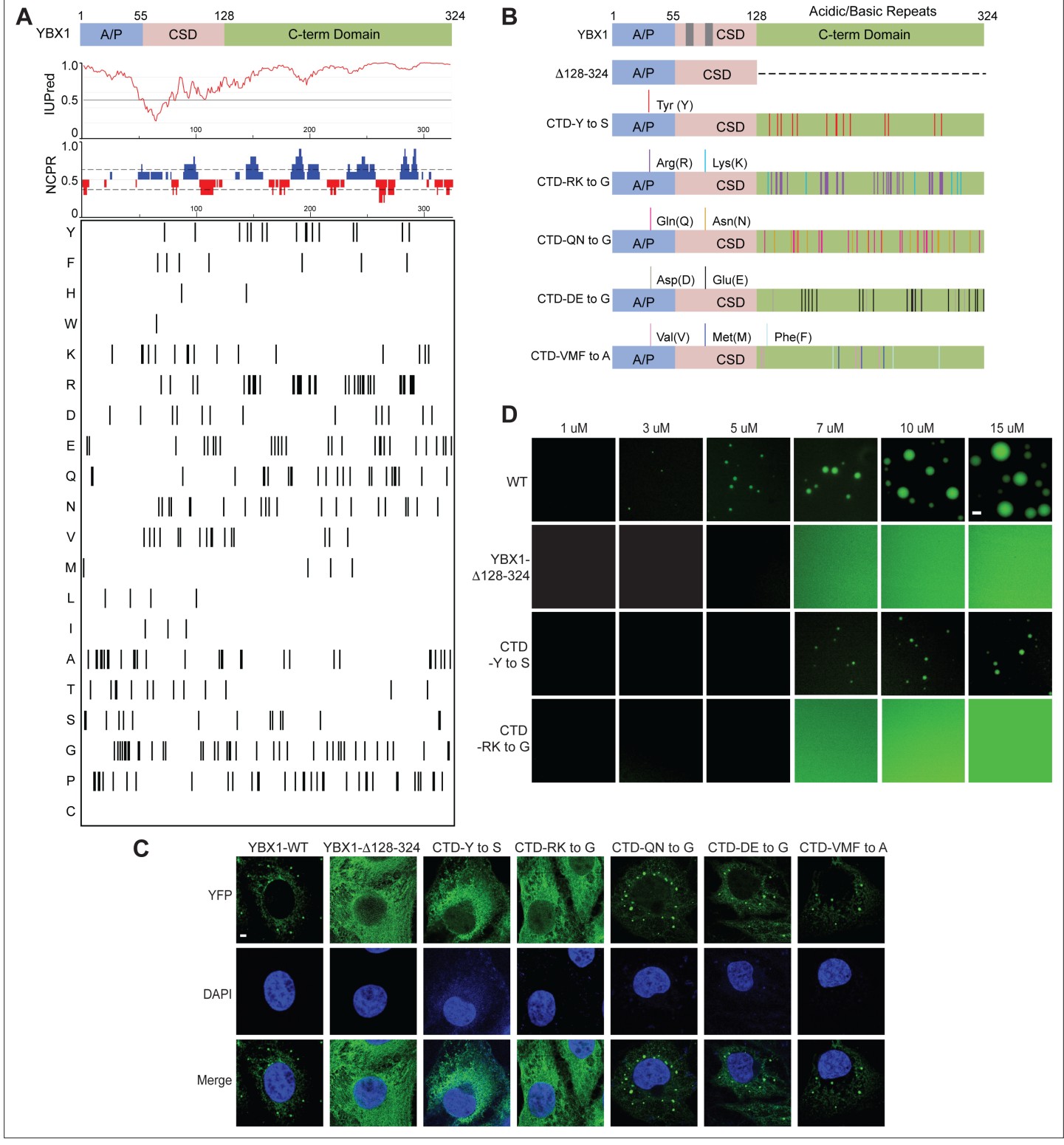

**Figure 3.** YBX1 phase separation is governed by association of aromatic and basic amino acids in C-terminal IDR. (**A**) Structural organization of YBX1. Top, IUPred, prediction of disordered protein regions; Middle, NCPR, net charge per residue with a sliding window of five residues; Net positive, blue, net negative, red; Bottom, visualization outputs for residue plots. (**B**) Schematic diagrams of different YBX1 mutants with the distribution of mutated amino acids. (**C**) Truncation mapping and identification of residues in YBX1 C-terminal IDR that are required for YBX1 condensation formation. YFP fused YBX1 wild type and mutants were introduced in ΔYBX1 U2OS cells by transient transfection and visualized by fluorescence microscopy. (**D**) Phase separation of YBX1 wild type and variants at the indicated concentrations. 6xHis-MBP-mGFP fused YBX1 wild type and variant proteins were purified

*Figure 3 continued on next page*

Figure 3 continued

from insect cells. Phase separation was induced by diluting the salt concentration from 500 mM to 150 mM in this assay. Scale bars, 3 µm.

The online version of this article includes the following figure supplement(s) for figure 3:

**Figure supplement 1.** YBX1 amino acid sequences and secondary structure prediction.

**Figure supplement 2.** The ability of YBX1 to form LLPS requires C-terminal IDR, likely depending on tyrosine and basic amino acids arginine and lysine.

**Figure supplement 2—source data 1.** Uncropped SDS-PAGE corresponding to **Figure 3**-figure supplement 2.

**Figure supplement 3.** A F85A mutation did not affect YBX1 liquid droplet formation *in vitro*.

YBX1 (**Figure 3A**). To further understand which amino acids contribute to IDR-driven YBX1 LLPS, we made several distinct variants of YBX based on the residue composition within the C-terminal IDR. Several polar but uncharged amino acids variants (CTD-Y to S/A, CTD-QN to G/A), a basic amino acid variant (CTD-RK to G), an acidic amino acid variant (CTD-DE to G) and a hydrophobic non-polar amino acid variant (CTD-VMF to A) were expressed in ΔYBX1 cells (**Figure 3B** and **Figure 3—figure supplement 2C**). We first examined the subcellular localization of YFP-fusion forms of these variants in U2OS cells by transient transfection. All the mutants expressed similar or higher levels of the fusion protein compared to wild type (**Figure 3—figure supplement 2F**). Either replacing all the tyrosine residues in the C-terminal disordered region of YBX1 with serine residues/alanine residues (CTD-Y to S/A), or replacing all the arginine and lysine residues with glycine (CTD-RK to G), comprehensively impaired YBX1 condensate formation in cells (**Figure 3C** and **Figure 3—figure supplement 2C**). All the other mutants formed YBX1 condensates as efficiently as wild type (**Figure 3C**). We next sought to determine if variants CTD-Y to S and CTD-RK to G were defective in LLPS of YBX1 *in vitro* with pure mGFP-fusion proteins expressed and isolated from insect cells (**Figure 3—figure supplement 2D**). In agreement with condensate formation in cells, YBX1 phase separation was severely impaired as indicated by the formation of smaller droplets at a higher range of protein concentrations of CTD-Y to S variant or completely deficient by substitution of arginine and lysine residues with glycine in the CTD (CTD-RK to G) (**Figure 3D**). We further noticed that tyrosines in the CTD are relatively uniformly distributed (**Figure 3B**), consistent with a phase separation that fits the sticker-and-spacer model pattern of aromatic residues such as has been proposed for RNA-binding proteins (**Martin et al., 2020**). These results suggest that YBX1 phase separation is driven by the IDR region, most likely through interactions among tyrosine- and arginine-rich motifs.

The YBX1 CSD is highly evolutionarily conserved and contains two consensus ribonucleoprotein (RNP) 1 and 2 sequences. The aromatic amino acid residues Phe74, Phe85, and His87 within these motifs form a hydrophobic cluster on the protein surface that participates in DNA or RNA binding (**Kloks et al., 2002**; **Yang et al., 2019**). A single amino acid Phe to Ala mutation (F85A), was reported to block YBX1-specific RNA binding (**Lyons et al., 2016**). The alternating basic and acidic clusters of the CTD are implicated in nonspecific nucleic acid binding as well as protein-protein interaction (**Mordovkina et al., 2020**). In our previous work, YBX1 was identified in purified exosomes and found to be required for sorting miR-223 into exosomes. We hypothesized that there might be a direct interaction between YBX1 and miR-223. Purified recombinant YBX1 interacts directly with 5' fluorescently-labeled miR-223 and F85A YBX1 abolishes that interaction as quantified by an electrophoretic mobility shift assay (Ma, L. and RS, in preparation). Mutation of Phe85 to Ala caused YBX1 translocation into the nucleus, mainly the nucleolus, suggesting that the RNA binding might be responsible for YBX1 retention in the cytoplasm (**Figure 3—figure supplement 3A**). Because the F85A mutant protein mostly localizes to the nucleus, it was used as a control in the following assays. To test whether YBX1-F85A remained in a liquid-like state in cells, we performed FRAP experiments (**Figure 3—figure supplement 3C and D**). Nearly 90 % of the fluorescence signal was recovered within 30 s after photobleaching, suggesting that YBX1-F85A is highly dynamic in cells. We further found that F85A did not affect YBX1 phase separation *in vitro* (**Figure 3—figure supplement 3B**).

## IDR-driven YBX1 phase separation is required for sorting YBX1 into exosomes

Exosomes are produced by intraluminal vesicle budding from the limiting membrane of MVBs. We devised a method to visualize the delivery of YBX1 to the lumen of MVBs. Enlarged endosomes are observed in cells overexpressing a constitutively active mutant mCherry-Rab5$^{Q79L}$ (**Baietti et al.,**

*2012*). Confocal microscopy revealed that the exosome marker CD63 but not EGFP filled the lumen or uniformly distributed over the rim of enlarged endosomes that were encircled by mCherry-Rab5^Q79L (*Figure 4A*). We also observed a marked accumulation of YFP-YBX1 in the lumen of enlarged endosomes (*Figure 4B*). In contrast, the RNA-binding defective YBX1 mutant (F85A) and phase separation defective mutants (CTD-Y to S, CTD-RK to G) did not appear in the lumen of ILVs within enlarged endosomes (*Figure 4—figure supplement 1A*), implying a requirement for phase separation in the incorporation of YBX1 into ILVs.

In *Shurtleff et al., 2016*, we documented that YBX1 co-purified with CD63-positive EVs secreted from HEK293T cells. In subsequent work by *Temoche-Diaz et al., 2019*, we showed that CD63 defines a distinct pool of high buoyant density vesicles corresponding to exosomes. To test the presence of YBX1 in EVs from cultured U2OS cells, we examined the fractionation of extracellular YBX1 by differential centrifugation and found by immunoblot that endogenous YBX1 co-sedimented with multiple EV markers (*Figure 4C*). Overexpression of YFP-YBX1 in ΔYBX1 cells enhanced the secretion of sedimentable YBX1 (*Figure 4C*). Fivefold more YFP-YBX1 was detected in the sediment of culture medium from cells overexpressing wt compared to F85A mutant YBX1 fusion protein. IDR-defective YBX1 mutant proteins (CTD-Y to S, CTD-RK to G) were less efficiently packaged than wt YBX1 into extracellular vesicles (*Figure 4D*). To confirm that YBX1 resided inside the lumen of extracellular vesicles, we performed proteinase K protection assays on membranes in the high-speed pellet fraction. As *Figure 4E* shows, endogenous YBX1 was protected from proteinase K digestion in the absence but not in the presence of Triton X-100. ALIX, a cytosolic protein within exosomes, and Flotillin-2, a membrane protein anchored to the inner leaflet of EVs, served as positive controls that were also degraded only in the presence of detergent. CD9, a multi (putative four)-transmembrane protein with an extracellular loop recognized by CD9 antibody, was vulnerable to degradation independent of detergent. Similarly, YFP-tagged YBX1 from a high-speed pellet fraction was mostly resistant to proteinase K (*Figure 4F*). These results confirmed that YBX1 was packaged into exosomes secreted from U2OS cells. In contrast, the RNA-binding defective YBX1 mutant F85A and IDR-defective mutants (Y to S and RK to G) were significantly decreased in high-speed pellet fractions (*Figure 4C and D*).

Cells overexpressing YFP-YBX1 were used for further purification of EVs by buoyant density flotation (*Figure 4G*). Isolated vesicles from U2OS averaged around 130 nm in diameter as determined by nanoparticle tracking analysis (NTA) (*Figure 4H*, vesicles from HEK293T cells averaged around 100 nm in diameter, *Figure 4—figure supplement 1B*). Vesicles examined by negative stain electron microscopy displayed a characteristic cup-shape (*Figure 4—figure supplement 1C*). YFP-YBX1 was detected in the buoyant vesicle fraction from ΔYBX1/YFP-YBX1 cells but not from ΔYBX1 cells (*Figure 4I*). Further separation of these vesicles was achieved on a linear iodixanol gradient (5–25%) which resolved two distinct EV species: low density (LD) and high density (HD) sub-populations, as we previously reported for EVs from MDA-MB-231 cells (*Temoche-Diaz et al., 2019*; *Figure 4J and K*). The YBX1 signal was detected in the combined HD vesicles which coincided with the exosome markers CD63 and ALIX (*Figure 4K*). Approximately 10-fold more YFP-YBX1 than YFP-YBX F85A mutant protein was detected in the HD vesicle fractions normalized to CD9 content in each (*Figure 4K*). These data illustrate that YBX1 sorting into exosomes is dependent on IDR-driven phase separation.

## IDR-driven YBX1 phase separation is required for sorting miRNA into exosomes

To test whether YBX1 condensation correlated with exosomal RNA sorting in cells, we reexamined the YBX1-dependent enrichment of miR-223 in exosomes purified by buoyant density flotation, as described in *Figure 4G*, from two different cell lines, HEK293T and U2OS. As before, we found that miR-223 and miR-144 were significantly enriched whereas cytoplasmic miR-190 was not enriched in purified exosomes compared to cells (*Figure 5A*; *Shurtleff et al., 2016*). Overexpression of YBX1 increased the relative miR-223 level in exosomes in both cell lines (*Figure 5B*). To confirm the requirement of YBX1 in sorting miR-223 into exosomes, we generated a YBX1 knockout HEK293T cell line with CRISPR/Cas9. YBX1 knockout clones (ΔYBX1-9, and ΔYBX1-41) were confirmed by Sanger sequencing for target DNA (*Figure 5—figure supplement 1A*), RT-qPCR for mRNA (*Figure 5—figure supplement 1B*) and immunoblot for YBX1 protein (*Figure 5C*). A similar knockout was made with U2OS cells (*Lyons et al., 2016*). We used RT-qPCR to quantify miR-223 levels in cells and exosomes, purified as described in *Figure 4G*. MiR-223 secretion into the growth medium and in isolated exosomes was

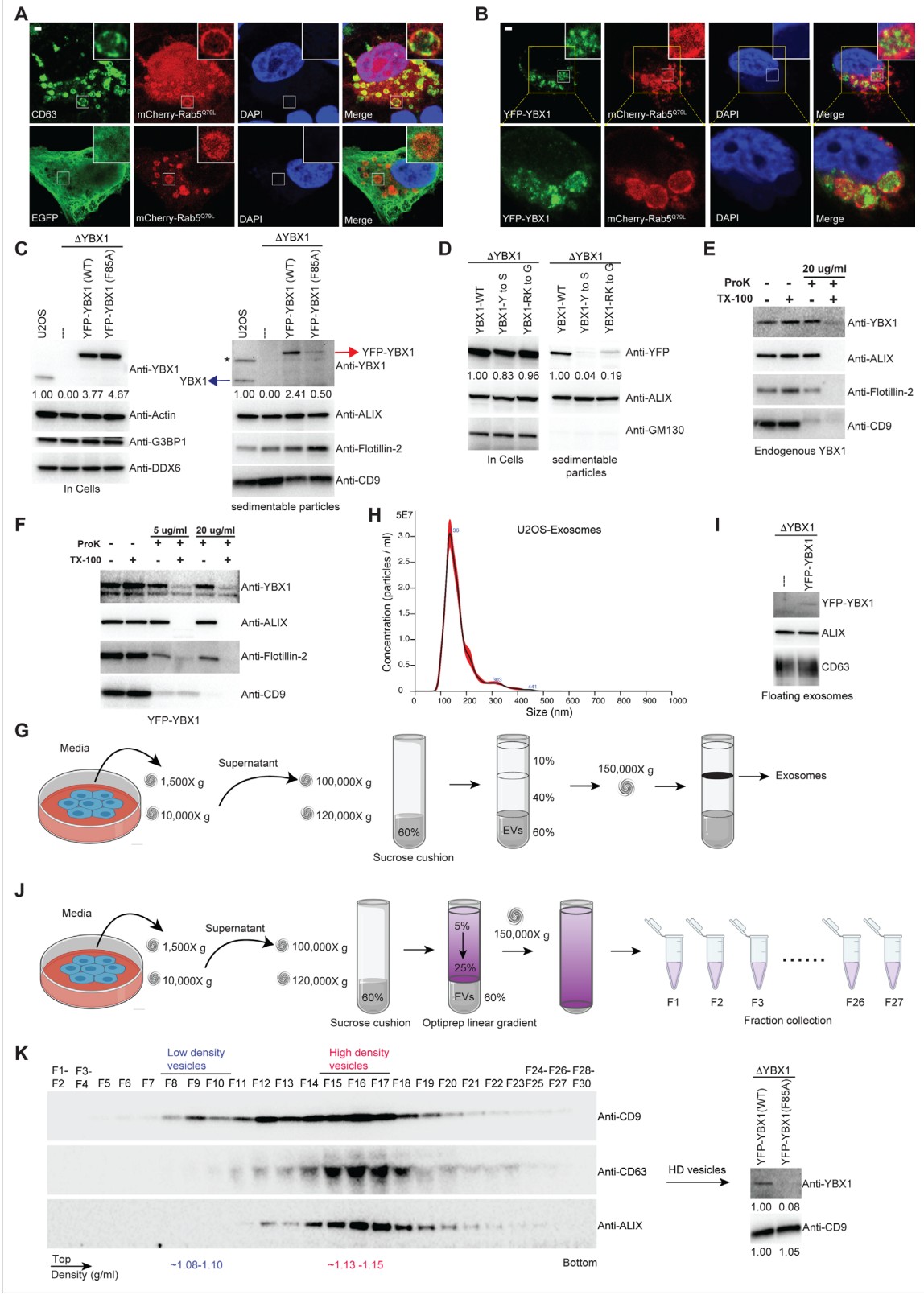

**Figure 4.** IDR-driven YBX1 phase separation is required for sorting YBX1 into exosomes. (**A**) Representative microscope images from U2OS cells expressing mChery-RAB5[Q79L]. Confocal micrographs of cells expressing mCherry-RAB5[Q79L], alone (upper row) or with EGFP (lower row). Cells are stained with anti-CD63 (upper row) or with anti-GFP (lower row). (**B**) Confocal micrographs of U2OS cells expressing mChery-RAB5[Q79L] and YFP-YBX1. (**C**) Over-expression of YBX1 in U2OS cells increased the secretion of YBX1 in EVs. Immunoblots for the indicated protein markers in cells and high-

*Figure 4 continued on next page*

*Figure 4 continued*

speed pellet fractions. The numbers under the YBX1 blot represent quantification analysis of endogenous YBX1, YFP-YBX1, and YFP-YBX1-F85A in cells and sedimentable particles by Fiji software. '*' is a non-specific band; Blue arrow represents endogenous YBX1; Red arrow represents fusion YBX1 or YBX1-F85A. (**D**) IDR-driven YBX1 phase separation is required for YBX1 secretion in EVs. Immunoblots for the indicated protein markers in U2OS cells and high-speed pellet fractions. The numbers under the YFP blot represent quantification analysis of endogenous YBX1 and variants in cells and sedimentable particles by Fiji software. (**E**) Proteinase K protection assay on high-speed pellet fractions from U2OS cells. Triton X-100 (0.5%) was used to disrupt the membranes. Immunoblots for YBX1, ALIX, Flotillin-2, and CD9 are shown. (**F**) Proteinase K protection assay on high-speed pellet fractions from U2OS cells expressing YFP-YBX1. Triton X-100 (0.5%) was used to disrupt the membranes. Immunoblots for YBX1, ALIX, Flotillin-2, and CD9 are shown. (**G**) Schematic showing exosome purification with buoyant density flotation in a sucrose step gradient. (**H**) Nanoparticle tracking analysis (NTA) quantification of exosomes from cultured U2OS cells. (**I**) YFP-YBX1 detected in sucrose post-flotation fraction from U2OS cells. Immunoblots for YBX1, ALIX, and CD63 from buoyant exosomes are shown. (**J**) Schematic showing exosome purification with buoyant density flotation in a linear iodixanol gradient. (**K**) Immunoblots across the iodixanol gradient from U2OS cells for classical exosome markers CD9, CD63 and ALIX (the left panel). Collection of fractions F15-F17 corresponding to high density vesicles and immunoblots for YBX1 and CD9. The numbers under YBX1 blot and CD9 blot represent quantification analysis of YFP-YBX1-WT or YFP-YBX1-F85A and CD9 in HD vesicles, respectively, by Fiji software. Scale bars, 3 µm.

The online version of this article includes the following figure supplement(s) for figure 4:

**Source data 1.** Uncropped Western blot images corresponding to **Figure 4C**.

**Source data 2.** Uncropped Western blot images corresponding to **Figure 4D**.

**Source data 3.** Uncropped Western blot images corresponding to **Figure 4E**.

**Source data 4.** Uncropped Western blot images corresponding to **Figure 4F**.

**Source data 5.** Uncropped Western blot images corresponding to **Figure 4I**.

**Source data 6.** Uncropped Western blot images corresponding to **Figure 4K**.

**Figure supplement 1.** YBX1 entering into ILVs is dependent on IDR-driven phase separation.

reduced ~2 fold and correspondingly accumulated within ΔYBX1 mutant derivatives of HEK293T and U2OS cells (*Figure 5D* and *Figure 5—figure supplement 1C*). Finally, to test the role of the YBX1 CTD in sorting of miR-223 into exosomes, we overexpressed YBX1-F85A, YBX1-Y to S and YBX1-RK to G mutants in ΔYBX1 U2OS cells and found similar miR-223 reductions in exosomes and accumulation within cells in all three mutant lines (*Figure 5E*). Similar results were seen for secretion of miR-223 into the medium fraction of ΔYBX1 mutant 293T cells overexpressing CTD mutants of YBX1 (*Figure 5—figure supplement 1D*). These results suggest that YBX1 condensation helps to recruit miR-223 for sorting into exosomes.

To test whether RNA regulates the phase behavior of YBX1, we performed a phase separation assay with recombinant mGFP-YBX1 in the presence of total RNA isolated from U2OS cells. We mixed increasing quantities of RNA with a fixed concentration of YBX1 and imaged the resulting droplets. Consistent with a previous report on prion-like RNA-binding protein FUS (*Maharana et al., 2018*), we found that increasing the RNA/YBX1 ratio initially promoted liquid droplet size until a point where droplets were less stable or were not produced (*Figure 6—figure supplement 1A*).

Given the cellular function of YBX1 involves sorting miR-223 into exosomes, we examined miR-223 capture into YBX1 droplets. Cy5(5')-labeled miR-223 was incubated with mGFP-YBX1 under phase separation conditions and observed by fluorescence microscopy. As shown in *Figure 6A*, miR-223 accumulated in liquid-like droplets coincident with YBX1.

We sought to identify the domains of YBX1 that contribute to the recruitment of miR-223. Cy5 (5')-labeled miR-223 was mixed with different YBX1 variants as shown in *Figure 6B*. Disrupting the association of YBX1 and miRNA through mutation of Phe85 to Ala had no effect on YBX1 droplet formation, but almost completely blocked miR-223 recruitment (*Figure 6B and C*). The YBX1-CTD-Y to S mutant greatly reduced the formation of droplets but those that formed recruited miR-223 at a similarly reduced level (*Figure 6B and C*). miR-223 condensation was not detected when YBX1 phase separation was completely blocked in the YBX1-CTD-RK to G mutant (*Figure 6B and C*). These data suggest that YBX1 recruits miR-223 through direct interaction with the central, cold shock domain, and into condensates governed by the C-terminal domain.

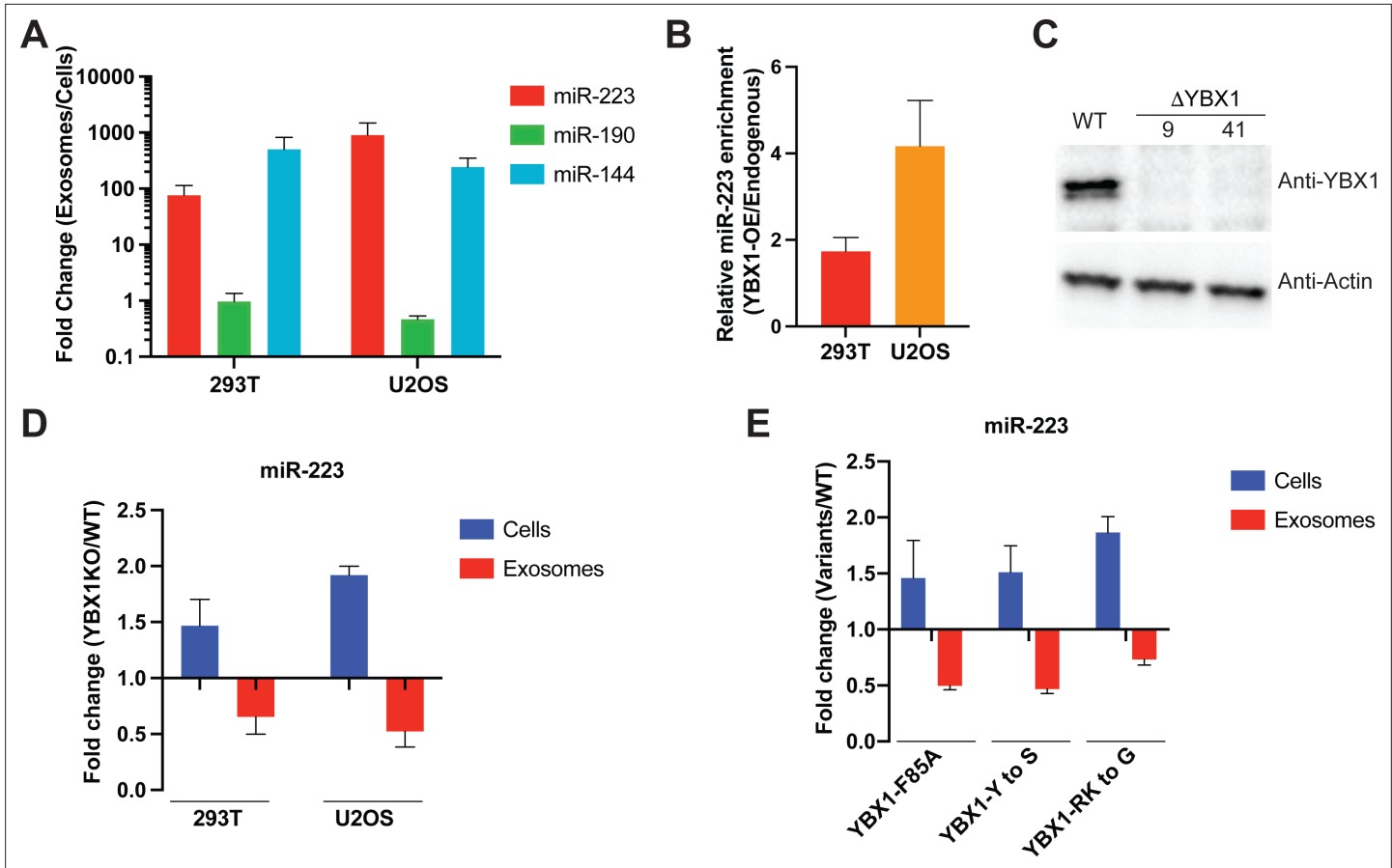

**Figure 5.** IDR-driven YBX1 phase separation is required for sorting miR-223 into exosomes. (**A**) Relative abundance of miRNAs detected in exosomes compared to cellular levels from both HEK293T cells and U2OS cells. Exosomes were purified as in *Figure 4G*. Fold change of miRNAs in cells and purified exosomes from indicated cells quantified by RT-qPCR. Data are plotted from three independent experiments and error bars represent standard derivations. (**B**) Overexpression of YBX1 increases sorting of miR-223 into exosomes both in HEK293T cells and U2OS cells. Exosomes were purified as in *Figure 4G*. Fold change of miR-223 in cells and purified exosomes from indicated cells quantified by RT-qPCR. Relative miR-223 enrichment was calculated by fold change (Exo/cells) of YBX1-OE divided by fold change (Exo/cells) of endogenous YBX1. Data are plotted from three independent experiments and error bars represent standard derivations. (**C**) Analysis of wild-type and CRISPR/Cas9 genome edited HEK293T clones by immunoblot for YBX1 (top) and actin (bottom). (**D**) The accumulation of miR-223 in cells and depletion of miR-223 in exosomes derived from ΔYBX1 and WT cells. Exosomes were purified as in *Figure 4G*. Fold change of miR-223 in cells and purified exosomes from indicated cells quantified by RT-qPCR. Data are plotted from three independent experiments for HEK293T cells and two independent experiments for U2OS cells; error bars represent standard derivations from independent samples. (**E**) Residues contributing to YBX1 phase separation are required for sorting miR-223 into exosomes. Exosomes were purified from U2OS cells as in *Figure 4J*. Fold change of miR-223 in cells and purified exosomes from indicated cells quantified by RT-qPCR. All quantifications represent means from three independent experiments and error bars represent standard derivations.

The online version of this article includes the following figure supplement(s) for figure 5:

**Source data 1.** Uncropped Western blot images corresponding to **Figure 5C**.

**Source data 2.** The numerical data that are represented as graphs in **Figure 5**.

**Figure supplement 1.** IDR-driven YBX1 phase separation is required for sorting miR-223 into the growth medium.

## YBX1 condensates recruit miRNAs and sort them into exosomes with selectivity

To examine whether YBX1 recruits miRNAs with selectivity, we analyzed two additional miRNAs: miR-190, an abundant cellular miRNA and miR-144, one that is highly enriched in exosomes (*Shurtleff et al., 2016*). We first incubated 5'Cy5-labeled miR-223, miR-190 and miR-144 with mGFP-YBX1 independently. Incubations were conducted in the presence of an excess of unlabeled RNA (10 ng/μl total RNA was extracted from U2OS cells) which produced a clear discrimination between miR-223 which partitioned well into YBX1 condensates from miR-190 and miR-144, which did not (*Figure 6D*

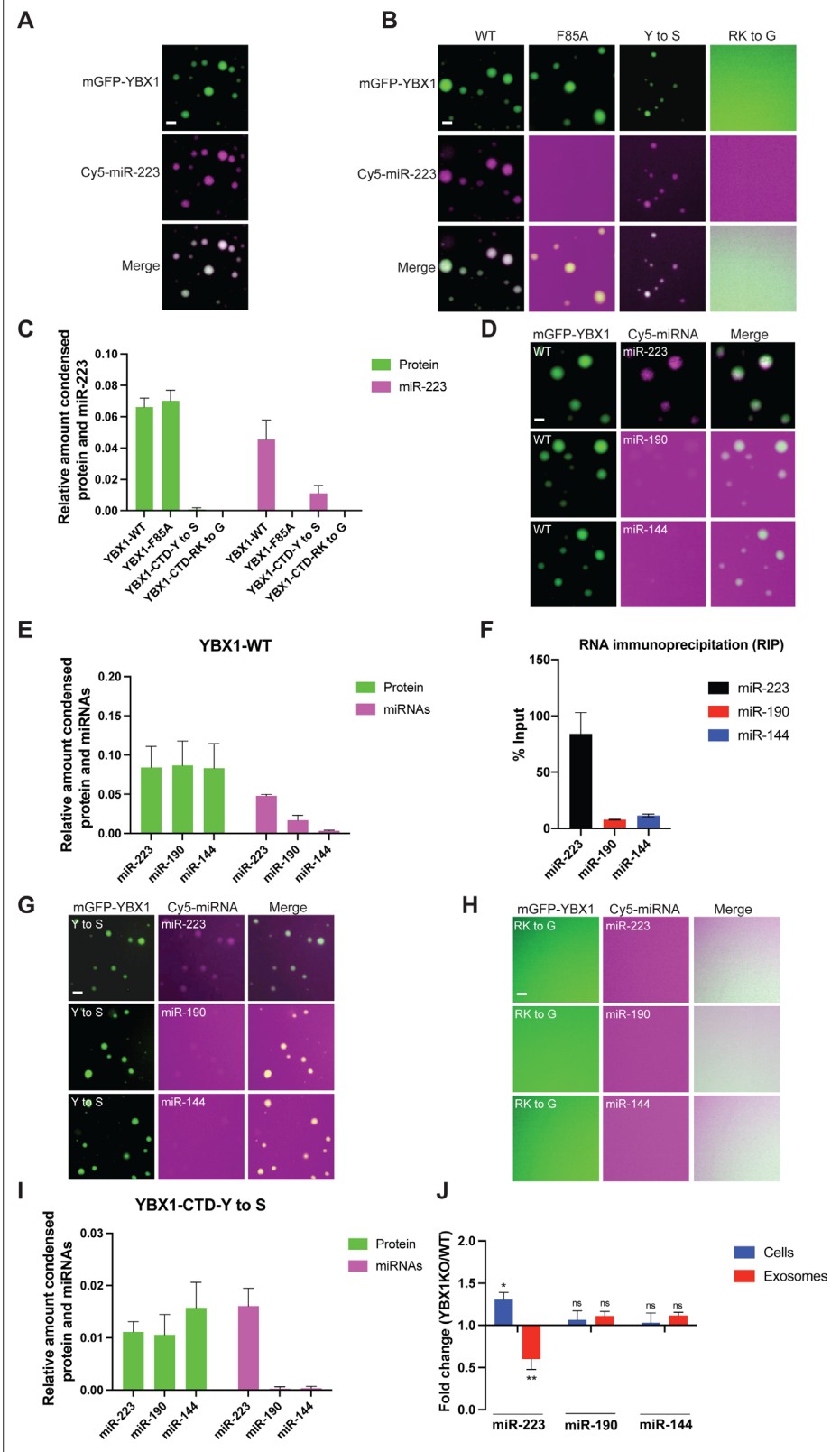

**Figure 6.** YBX1 phase-separated droplets recruit miRNAs with selectivity correlated with the exosome sorting ability *in vivo*. (**A**) YBX1 phase-separated droplets recruit miR-223. Purified mGFP-YBX1 was incubated with Cy5 labeled miR-223 together with 10 ng/µl total RNA in LLPS buffer and then observed under a microscope. (**B, C**) The recruitment of miR-223 into YBX1 phase-separated droplets depends on the ability of YBX1 to bind RNA

*Figure 6 continued on next page*

*Figure 6 continued*

rather than phase separation. Representative images (400 × 400 pixels) (**B**) and quantification (**C**) of condensed miR-223 and YBX1 protein. Relative amount condensed protein or miRNAs was calculated as ratio of total intensity of protein inside droplets to total intensity of protein both inside and outside of droplets as quantified using Fiji software. Three images (2048 x 2048 pixels) for per condition were analyzed. The results are plotted as the mean ± the standard deviation (SD). (**D, E**) YBX1 liquid droplets recruit miRNAs in a selective manner. Purified mGFP-YBX1 was incubated with Cy5 labeled miR-223, miR-190, or miR-144 individually, together with 10 ng/μl total cellular RNA in LLPS buffer and then observed under a microscope. Representative images (400 × 400 pixels) (**D**) and quantification (**E**) of condensed miRNAs and YBX1 protein. Relative amounts of condensed protein and RNA were calculated as described in (**C**). Three images (2048 x 2048 pixels) per condition were analyzed. The results are plotted as the mean ± the standard deviation (SD). (**F**) RIP assay with GFP-trap beads on YFP-YBX1 expressing HEK293T cell extracts. miRNAs in immunoprecipitated samples were determined by RT-qPCR using Taqman miRNAs assay, and reported as percentage of input sample (% input). Data are plotted as means ± SD of three independent experiments. (**G**) YBX1-CTD-Y to S mutant recruits miRNAs inefficiently but selectively. Purified mGFP-YBX1-CTD-Y to S was incubated with Cy5 labeled miR-223, miR-190 or miR-144 independently, together with 10 ng/μl total cellular RNA in LLPS buffer and then observed under a microscope. (**H**) YBX1-CTD-RK to G mutant failed to phase separate and recruit miRNAs. Purified mGFP-YBX1-CTD-RK to G was incubated with Cy5 labeled miR-223, miR-190 or miR-144 independently, together with 10 ng/μl total cellular RNA in LLPS buffer and then observed under a microscope. (**I**) Quantification of condensed miRNAs and YBX1 protein from (**G**). Relative amount of condensed protein and RNA were calculated as described in *Figure 6C*. Three images (2048 x 2048 pixels) per condition were analyzed. The results are plotted as the mean ± the standard deviation (SD). (**J**) YBX1 is required for sorting miR-223 but not miR-190 and miR-144 into exosomes. Exosomes were purified as in *Figure 4G*. Fold change of miR-223, miR-190, and miR-144 in cells and purified exosomes from indicated cells quantified by RT-qPCR. All quantifications represent means from three independent experiments and error bars represent standard derivations. Statistical significance was performed using unpaired t-test (*p < 0.05, **p < 0.01, and ns = not significant). Scale bars, 3 μm.

The online version of this article includes the following figure supplement(s) for figure 6:

**Source data 1.** The numerical data that are represented as graphs in **Figure 6.**

**Figure supplement 1.** RNA regulates the phase separation behavior of YBX1.

**Figure supplement 2.** miRNAs differ in affinity to YBX1 phase-separated droplets.

**Figure supplement 3.** YBX1 condensation is required for sorting miR-223 but not miR-190 and miR-144 into exosomes.

*and E*). The addition of unlabeled excess RNA produced enlarged YBX1 droplets (*Figure 6—figure supplement 2A*). At a fixed concentration of 7.5 μM YBX1 and 100 nM miRNA but without unlabeled excess RNA, both miR-223 and miR-190 partitioned into YBX1 droplets (*Figure 6—figure supplement 2B and C*). However, at varied concentrations of miRNA, the partition coefficient for miR-223 into YBX1 droplets was higher than miR-190, while the partition coefficients for YBX1 were almost same (*Figure 6—figure supplement 2C*). Additional tests of selectivity were conducted in the presence of unlabeled cellular RNA.

Unlike miR-190, miR-144 was detected enriched in EVs and dependent on YBX1 for secretion from HEK293T cells (*Shurtleff et al., 2016*). Nonetheless, miR-144 was not recruited in YBX1 droplets (*Figure 6D and E*). In an independent assay, we evaluated the interaction of YBX1 with three miRNAs by co-immunoprecipitation from HEK293T cell lysates. These results were consistent with the capture or not of these miRNAs in YBX1 condensates with nearly quantitative co-precipitation of YBX1 and miR-223 but 5-10-fold lower co-precipitation of miR-190 and miR-144 from cell lysates (*Figure 6F*). We next assessed miRNA partition into the phase separation defective mutants YBX1-CTD-Y to S and YBX1-CTD-RK to G (*Figure 6G, H and I*). The Y to S mutant produced less condensate at the same protein concentration and was somewhat less discriminatory and the RK-G mutant produced no visible condensate of protein or RNA. Although the results of these two experiments were consistent, the requirement of YBX1 for secretion of miR-144 in cells was not reflected in a requirement for capture by pure YBX1 protein in condensates.

In our previous work, we evaluated the requirement for YBX1 in the secretion of miR-223 and miR-144 by assaying samples of the culture medium in which HEK293T cells were grown. We sought to refine this measurement by quantifying the miRNA content of buoyant vesicles secreted in the culture medium as we did for the experiment in *Figure 5D*. As shown in *Figure 5A*, among these

three miRNAs, miR-223, and miR-144 but not miR-190 were enriched to different extents in exosomes secreted by HEK293T and U2OS cells. Sorting of miR-223 but not miR-190 and miR-144 into exosomes was decreased in ΔYBX1 cells (*Figure 6J*) and IDR defective mutants (*Figure 6—figure supplement 3*). Combined with the other results of our current work, we conclude, as before, that YBX1 enhances the secretion of miR-223 in EVs but that miR-144, although enriched in EVs, does not engage YBX1 condensates and is not required for secretion in EVs. It seems likely that another RBP is responsible for sorting of miR-144 into exosomes.

## Condensation of YBX1 into P-bodies is required for sorting miRNAs into exosomes

YBX1 was previously suggested to be a component of P-bodies that form foci together with Dcp1a as visualized by fluorescence microscopy (*Yang and Bloch, 2007*). We observed endogenous YBX1 colocalized with P-body components, EDC4, Dcp1a, and DDX6, as visualized by specific antibodies (*Figure 7A and B*). To address the role of condensation in the incorporation of YBX1 in P-bodies, we analyzed the co-localization of IDR mutants with EDC4. Both YBX1-CTD-Y to S and YBX1-CTD-RK to G mutants largely eliminated YBX1 condensation (*Figure 7C*).

To further study the association of YBX1 and P-bodies, we performed affinity purification coupled with mass spectrometry analysis of N-terminally EGFP-tagged or 3xFlag-tagged YBX1. Comparing positive hits found with both tagged forms of YBX1 immunoprecipitation trials, we generated a proteome of potential YBX1 interactors and compared this with a published P-body proteome (*Hubstenberger et al., 2017*; *Figure 7—source data 1*, *Figure 7D*). About 35 % (43/125) of P-body proteins were identified as potential YBX1 binding partners. Gene Ontology analysis showed that RNA-binding proteins were enriched in the YBX1 interactome (*Figure 7E*). Some of the RNA-binding proteins, such as SYNCRIP and SSB (Lupus La protein), were identified previously for roles in sorting miRNAs into exosomes (*Santangelo et al., 2016*; *Temoche-Diaz et al., 2019*). We observed that SYNCRIP formed condensates and co-localized with YBX1 (*Figure 7—figure supplement 1A*) in cells, implying that the condensation properties might be shared by other RBPs that are involved in exosomal RNA sorting. The potential YBX1 interactors included components of the miRNA processing pathway, MOV10 and Ago2, and well-known P-body markers, DDX6 and EDC4 (*Figure 7F*).

DDX6, a DEAD box helicase, plays a key role in P-body assembly, and interacts with almost half of P-body proteins (*Hubstenberger et al., 2017*). Yeast Dhh1 (human DDX6) undergoes LLPS *in vitro* and controls processing body dynamics *in vivo* through its RNA-stimulated ATPase activity (*Mugler et al., 2016*). Using co-immunoprecipitation, we found that wild type but not condensation-defective mutant forms of YBX1 interacted with DDX6 (*Figure 7G and H*). Correspondingly, we found that DDX6 was sorted into the luminal interior of isolated EVs as judged by buoyant density fractionation and a proteinase k protection assay (*Figure 7I*). To extend this analysis, we purified EVs from HEK293T cells by buoyant density flotation as described in *Figure 4G* and conducted a proteomic analysis using liquid chromatography tandem mass spectrometry (LC-MS/MS) (*Figure 7—source data 2*). Compared with the published P-body proteome, we found that 18.4 % (23/125) of P-body proteins were identified in exosomes (*Figure 7J* and *Figure 7—source data 2*). Treatment of cells with the translational inhibitor CHX (cycloheximide) is well established to trap mRNAs in polysomes and prevent P-body assembly (*Cougot et al., 2004*). To test whether P-bodies are involved in exosomal miRNA sorting, we treated U2OS cells with 10 µg/ml CHX for 2 hr and found that both the P body marker EDC4 and YBX1 were no longer observed in foci (*Figure 7—figure supplement 2A*). Exosome number but not particles quantified from the high-speed pellet fraction declined after CHX treatment as determined by nanosight nanoparticle tracking analysis (*Figure 7—figure supplement 2C*). Correspondingly, miR-223 secretion into the medium was significantly reduced (*Figure 7—figure supplement 2B*) and declined two fold in exosomes (*Figure 7—figure supplement 2D*) as determined by RT-qPCR. Thus, there may be a role for P-bodies in the concentrative capture of proteins destined for secretion in EVs.

## Discussion

Several RNA-binding proteins (RBPs) involved in the sorting of miRNAs into exosomes secreted by mammalian cells share a sequence domain, the IDR, implicated in the association of RNA and proteins in membraneless organelles such as P-bodies (*Lee et al., 2020*; *Luo et al., 2018*; *Santangelo et al.,*

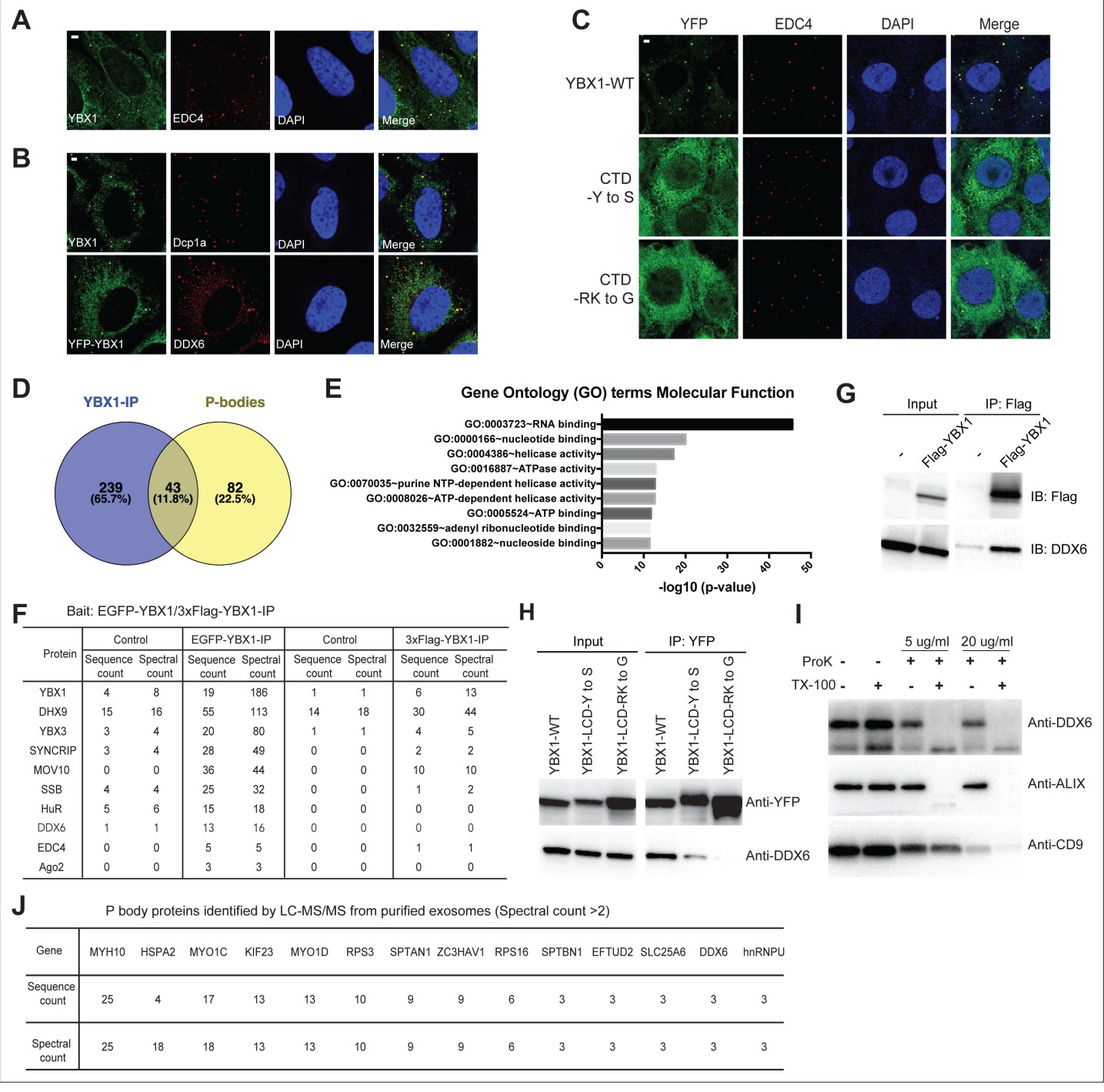

**Figure 7.** Condensation of YBX1 in PBs is required for sorting miRNAs into exosomes. (**A**) YBX1 condensates co-localized with P-body marker EDC4. Indirect immunofluorescence was used to show that YBX1 localized to P-bodies. U2OS cells were stained with anti-YBX1 and anti-EDC4 antibodies. (**B**) YBX1 condensates co-localized with P-body markers Dcp1a and DDX6. U2OS cells were stained with anti-YBX1 and anti-Dcp1 antibodies (upper row), or with anti-YFP and anti-DDX6 antibodies (lower row). (**C**) YBX1 condensation into P-bodies dependent on IDR-driven phase separation. YFP-fused YBX1 wild type and variants were introduced in ΔYBX1 U2OS cells by stable transfection and visualized by fluorescence microscopy. Cells were stained with anti-YBX1 and anti-EDC4 antibodies. (**D**) The Venn diagram shows overlap between YBX1 proteome and previously reported P-body proteome. (**E**) GO analysis (molecular function) of genes associated with YBX1. (**F**) Proteins identified by either 3xFlag-YBX1-IP or mGFP-YBX1-IP, coupled with mass spectrometry. (**G**) Coimmunoprecipitation of DDX6 with YBX1 from HEK293T cells. (**H**) Residues in YBX1-IDR that drive LLPS are required for its interaction with DDX6 in HEK293T cells. (**I**) DDX6 resides in exosomes. Proteinase K protection assay for DDX6 using exosomes that were isolated by buoyant density flotation from HEK293T cells. Triton X-100 (0.5%) was used to disrupt the membranes. Immunoblots for DDX6, ALIX, and CD9 are shown. (**J**) Identification of P-body components in purified exosomes from HEK293T cells by LC-MS/MS. Exosomes were purified as in *Figure 4G*. Scale

*Figure 7 continued on next page*

*Figure 7 continued*

bars, 3 µm.

The online version of this article includes the following figure supplement(s) for figure 7:

**Source data 1.** The proteins identified by YBX1 immunoprecipitation coupled with mass spec analysis and P bodies proteome.

**Source data 2.** The proteins identified in exosomes from HEK293T cells and P bodies proteome.

**Source data 3.** Uncropped Western blot images corresponding to **Figure 7G**.

**Source data 4.** Uncropped Western blot images corresponding to **Figure 7H**.

**Source data 5.** Uncropped Western blot images corresponding to **Figure 7I**.

**Source data 6.** The proteins identified in exosomes from HEK293T cells and stress granules proteome.

**Figure supplement 1.** SYNCRIP forms condensates and co-localizes with YBX1 and P-body marker DDX6.

**Figure supplement 2.** Disruption of P-body formation by CHX decreases miR-223 sorting into exosomes.

**Figure supplement 3.** Effects of shRNA knockdown of selected P body proteins on miR-223 sorting into exosomes.

**Figure supplement 3—source data 1.** Uncropped Western blot images corresponding to **Figure 7-** figure supplement 3B.

**Figure supplement 3—source data 2.** Oligo sequences used for shRNA cloning for DDX6, 4E-T, and LSM14A.

*2016*; *Shurtleff et al., 2016*; *Temoche-Diaz et al., 2019*; *Xing et al., 2020*). We previously reported roles for the YBX1 protein in sorting and secretion of miR-223 in HEK293T cells and the Lupus La protein in secretion of miR-122 in MDA-MB-231 cells. Both proteins contain such IDR domains, and at least one, the La protein, appears to be organized in puncta in the cytoplasm of MDA-MB-231 cells (*Shurtleff et al., 2016*; *Temoche-Diaz et al., 2019*). Here, we report that pure YBX1 forms a liquid-like condensate and similarly associates with P-bodies in cells including in apparent association with other bona fide constituents of P-bodies such as the DDX6 protein. YBX1 also contains an RNA-binding domain, the CSD, which together with the IDR is required for sorting of miR-223 into exosomes secreted by cells and for the species-selective concentration of miR-223 into condensates *in vitro*. We report a previously unrecognized connection between proteins such as YBX1 and DDX6 that associate in P-bodies and are also secreted in exosomes. Thus, we speculate that the P-bodies may house miRNA and other small RNAs prior to their capture into membrane invaginations and intralumenal vesicles of the multivesicular body (MVB). Fusion of the MVB at lysosome would then subject the ILVs and their condensate content to degradation. Fusion of the MVB at the cell surface would result in the extracellular discharge of exosomes for uptake into other cells.

Several other RBPs have been reported to play a role in the sorting and secretion of miRNAs in exosomes. These include hnRNPA2B1 (*Villarroya-Beltri et al., 2013*), SYNCRIP (*Hobor et al., 2018*; *Santangelo et al., 2016*), HuR (*Mukherjee et al., 2016*), and the major vault protein (MVP) (*Statello et al., 2018*; *Teng et al., 2017*), one of which, SYNCRIP, contains an IDR domain and associates with YBX1 in P-bodies (*Figure 7—figure supplement 1A*). Other groups have developed evidence for post-translational modifications of RBPs that may influence their localization in cells. Sumoylation, uridylation and ubiquitylation of RBPs have been implicated in the selection of miRNA cargo for secretion in EVs (*Koppers-Lalic et al., 2014*; *Villarroya-Beltri et al., 2013*). The secretion of YBX1 in EVs, for example, has been shown to depend on ubiquitylation (*Palicharla and Maddika, 2015*). The relationship between these modifications and the capture of miRNAs into invaginations that form on an endosome remains to be explored. On possible connection is in the role of post-translational modifications such as phosphorylation, methylation, and ubiquitylation of IDRs where the phase transition properties are altered by changes in charge, hydrophobicity size and structure (*Owen and Shewmaker, 2019*). Such modifications may reversibly control the partition of RBPs and cargo into RNA granules and subsequently into vesicles budding into the endosome.

Of the two IDRs in YBX1, only the one located in the C-terminal domain appears to influence the phase condensation properties of the protein. Within the C-terminal IDR, we identified Y, R, and K residues that contribute to LLPS. As shown for prion-like RNA binding properties, we suggest that phase separation of YBX1 is governed by interactions between Y and R residues (*Wang et al., 2018*). RNA-binding through an interaction with the CSD domain of YBX1 appears to reinforce phase separation producing larger droplets at an optimum ratio of RNA/protein (*Figure 6—figure supplement 1A*).

Dissolution of P-bodies by CHX treatment reduced by two fold the sorting of miR-223 but not miR-144 into density gradient purified exosomes (*Figure 7—figure supplement 2*). CHX inhibits protein

synthesis by trapping mRNA on ribosomes and thereby prevents their influx into P bodies. Three proteins have been reported to be essential for P-body formation in mammalian cells: DDX6, 4E-T, and, to a lesser extent, LSM14A (*Ayache et al., 2015*; *Standart and Weil, 2018*). To examine the direct role of P bodies in exosomal RNA sorting, we knocked down individual P-body protein DDX6, 4E-T or LSM14A by shRNA. We observed that P-body formation was greatly disrupted in most of the knock- down cells (*Figure 7—figure supplement 3A*, **type A cells**) with a corresponding defect in the sorting of miR-223 into exosomes from 4E-T knockdown cells but not DDX6 or LSM14A knockdown cells (*Figure 7—figure supplement 3C*). This correlated with the incomplete knockdown of DDX6 and LSM14A (*Figure 7—figure supplement 3B*) as also seen in the persistence of small (type A cells) and big granules (type B cells) (*Figure 7—figure supplement 3A*).

YBX1 has also been shown to localize to stress granules. Compared with the published stress granule proteome (*Jain et al., 2016*), we found that 28.7 % (118/411) of stress granule proteins were identified in exosomes (*Figure 7—source data 3*). YBX1 enrichment in stress granules has been reported to be linked to its secretion in response to the stress induced by arsenite treatment (*Guarino et al., 2018*).

Our observation of a connection between P-bodies and RBPs engaged in sorting of miRNAs for secretion in exosomes may relate the two organelles by function. P-bodies are thought to function in mRNA storage or decay. Although exosomes have some small mRNAs, the average length of RNAs in the range of 100 nt favors smaller species such as tRNA, Y-RNA, vault RNA and miRNAs. If P-bodies or other RNA granules serve to condense these small RNAs for secretion in exosomes, there would have to be some segregation of species with respect to size and perhaps function. Furthermore, P-bodies are larger than the typical exosome (~500 nm vs 30–150 nm) (*Hubstenberger et al., 2017*), thus the sorting of small RNA cargo for secretion may occur in only a subset of smaller P bodies or by a physical segregation of a domain of the P body by tubulation or budding into an invagination into the endosome. A further speculative connection relates to the localization of miRNA processing components, such as Ago2 (*Gibbings et al., 2009*; *Siomi and Siomi, 2009*), on the surface of endosomes and on our observation of a physical contact between YBX1 and Ago2 and MOV10 (*Figure 7F*). Nonetheless, as we consistently find Ago2 not localized within exosomes (*Shurtleff et al., 2016*; *Temoche-Diaz et al., 2019*), there must be some sorting of YBX1 from other P body content in the capture of proteins and RNA segregated into membrane buds invaginating into the endosome. We suggest that molecules destined for secretion in exosomes are segregated in a two-step process involving partition into a precursor larger RNA granule or into RNA granules of distinct function followed by sorting from or among granules to capture those molecules fated for secretion in exosomes and capture by target cells into which exosomes are internalized (*Figure 8*).

# Materials and methods

**Key resources table**

| Reagent type (species) or resource | Designation | Source or reference | Identifiers | Additional information |
|---|---|---|---|---|
| Gene (*Homo-sapiens*) | YBX1 | Addgene | RRID:Addgene_19878 | |
| Cell line (*Homo-sapiens*) | U-2 OS cells | PMID:27174937 | | Gift of Dr. Pavel Ivanov lab |
| Cell line (*Homo-sapiens*) | U-2 OS ΔYBX1 cells | PMID:27174937 | | Gift of Dr. Pavel Ivanov lab |
| Cell line (*Homo-sapiens*) | HEK293T cells | Other | | Cell culture facility at UC Berkeley |
| Cell line (*Homo-sapiens*) | HEK293T ΔYBX1 cells | This study | | Obtained by CRISPR-Cas9, cell line maintained in Schekman lab |
| Cell line (*Homo-sapiens*) | U-2 OS cells | Other | | Cell culture facility at UC Berkeley |
| Cell line (*Spodoptera frugiperda*) | Sf9 cells | Other | | Cell culture facility at UC Berkeley |

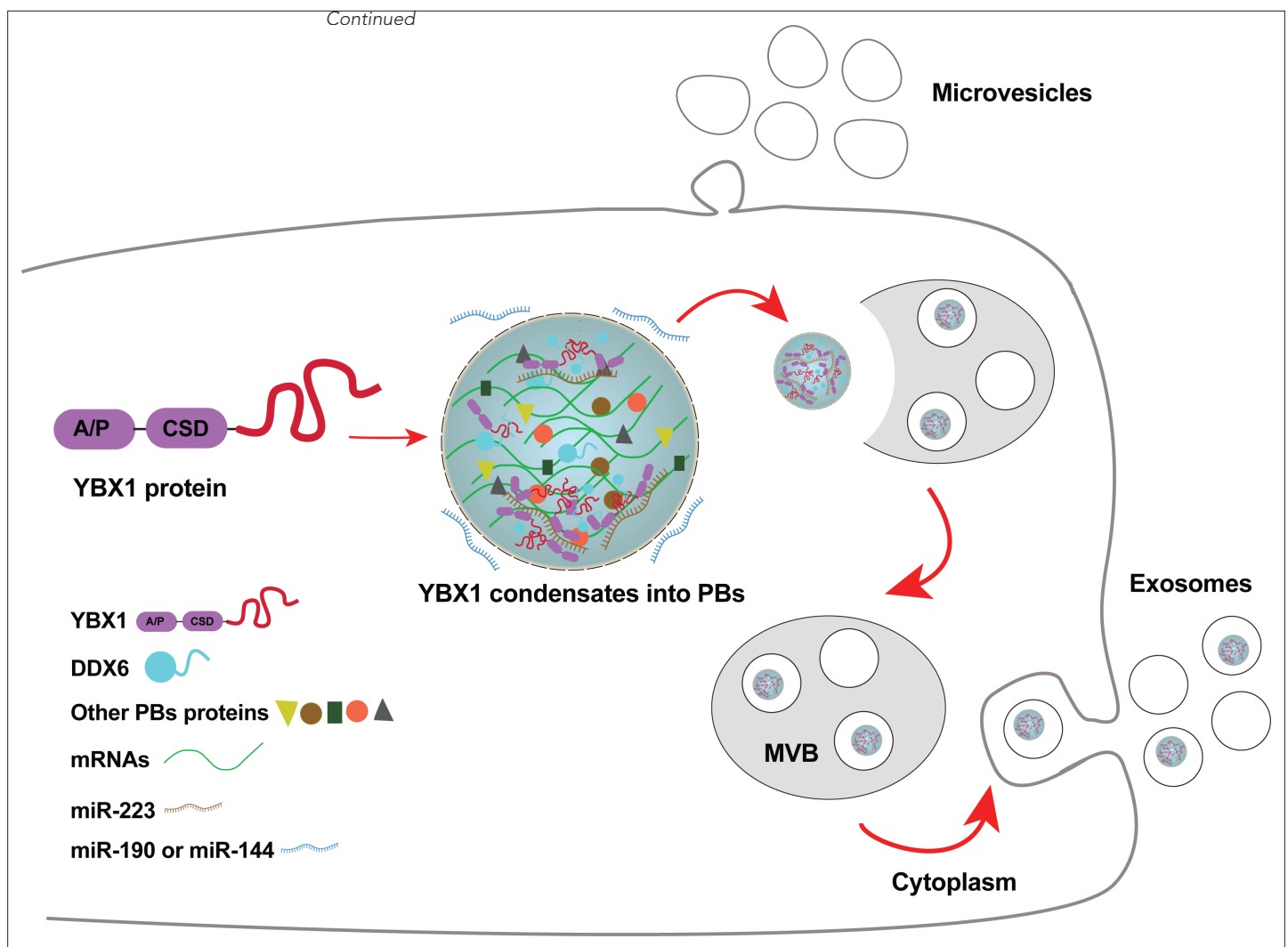

**Figure 8.** Diagram representing a working model of miRNA selectively sorted into exosomes by phase-separated YBX1 condensates. Cytosolic RBP YBX1 forms liquid-like condensates in cells and liquid droplets *in vitro*. Phase separation of YBX1 is governed by a C-terminal IDR, most likely through the association of aromatic amino acid tyrosine and basic amino acids arginine or lysine. Phase-separated YBX1 recruits miRNAs in a selective manner through N-terminal CSD- mediated specific protein-RNA interaction. YBX1 condensation increases its local concentration and the affiliation with P body components (such as DDX6), which further facilitates YBX1 and its cognate miRNAs sorting into exosomes. Segregation of RNA and RBPs for capture by invagination into an endosome occurs at the level of granule formation or by sorting of selected RNAs and RBPs from larger, more heterogeneous granules.

| Reagent type (species) or resource | Designation | Source or reference | Identifiers | Additional information |
|---|---|---|---|---|
| Recombinant DNA reagent | mCherry-Rab5CA(Q79L) (plasmid) | Addgene | RRID:Addgene_35138 | |
| Recombinant DNA reagent | EGFP-YBX1 (plasmid) | This study | | EGFP tagged YBX1, plasmid maintained in Schekman lab |
| Recombinant DNA reagent | 3xFlag-YBX1 (plasmid) | This study | | 3xFlag tagged YBX1, plasmid maintained in Schekman lab |
| Recombinant DNA reagent | YFP-YBX1 (plasmid) | *Continued on next page*  This study | | YFP tagged YBX1, plasmid maintained in Schekman lab |

*Continued*

| Reagent type (species) or resource | Designation | Source or reference | Identifiers | Additional information |
|---|---|---|---|---|
| Recombinant DNA reagent | YFP-YBX1-F85A (plasmid) | This study | | YFP tagged YBX1-F85A, plasmid maintained in Schekman lab |
| Recombinant DNA reagent | YFP-YBX1-(128-324)-Y to S/A (plasmid) | This study | | YFP tagged YBX1-(128-324)-Y to S/A, plasmid maintained in Schekman lab |
| Recombinant DNA reagent | YFP-YBX1-(128-324)-RK to G (plasmid) | This study | | YFP tagged YBX1-(128-324)-RK to G, plasmid maintained in Schekman lab |
| Recombinant DNA reagent | YFP-YBX1-(128-324)-QN to G/A (plasmid) | This study | | YFP tagged YBX1-(128-324)-QN to G/A, plasmid maintained in Schekman lab |
| Recombinant DNA reagent | YFP-YBX1-(128-324)-DE to G (plasmid) | This study | | YFP tagged YBX1-(128-324)-DE to G, plasmid maintained in Schekman lab |
| Recombinant DNA reagent | YFP-YBX1-(128-324)-VMF to A (plasmid) | This study | | YFP tagged YBX1-(128-324)-VMF to A, plasmid maintained in Schekman lab |
| Recombinant DNA reagent | YFP-YBX1-Δ (1-55) (plasmid) | This study | | YFP tagged YBX1-Δ (1-55), plasmid maintained in Schekman lab |
| Recombinant DNA reagent | YFP-YBX1-Δ (56-127) (plasmid) | This study | | YFP tagged YBX1-Δ (56-127), plasmid maintained in Schekman lab |
| Recombinant DNA reagent | YFP-YBX1-Δ (1-127) (plasmid) | This study | | YFP tagged YBX1-Δ (1-127), plasmid maintained in Schekman lab |
| Recombinant DNA reagent | YFP-YBX1-Δ (56-324) (plasmid) | This study | | YFP tagged YBX1-Δ (56-324), plasmid maintained in Schekman lab |
| Recombinant DNA reagent | YFP-YBX1-(56-127) (plasmid) | This study | | YFP tagged YBX1-(56-127), plasmid maintained in Schekman lab |
| Recombinant DNA reagent | YFP-YBX1-Δ (128-324) (plasmid) | This study | | YFP tagged YBX1-Δ (128-324), plasmid maintained in Schekman lab |
| Recombinant DNA reagent | His6-MBP-3C-mGFP-TEV-NotI-ccdB-AscI-stop-HindIII cassette (plasmid) | Addgene | RRID:Addgene_118890 | Gift of Dr. Anthony A. Hyman lab |
| Recombinant DNA reagent | His6-MBP-mGFP-YBX1 (plasmid) | This study | | To express YBX1 in insect cells, plasmid maintained in Schekman lab |

*Continued on next page*

Continued

| Reagent type (species) or resource | Designation | Source or reference | Identifiers | Additional information |
|---|---|---|---|---|
| Recombinant DNA reagent | His6-MBP-mGFP-YBX1-F85A (plasmid) | This study | | To express YBX1-F85A in insect cells, plasmid maintained in Schekman lab |
| Recombinant DNA reagent | His6-MBP-mGFP-YBX1-(128-324)-Y to S (plasmid) | This study | | To express YBX1-(128-324)-Y to S in insect cells, plasmid maintained in Schekman lab |
| Recombinant DNA reagent | His6-MBP-mGFP-YBX1-(128-324)-RK to G (plasmid) | This study | | To express YBX1-(128-324)-RK to G in insect cells, plasmid maintained in Schekman lab |
| Recombinant DNA reagent | His6-MBP-mGFP-YBX1-Δ (128-324) (plasmid) | This study | | To express YBX1-Δ (128-324) in insect cells, plasmid maintained in Schekman lab |
| Recombinant DNA reagent | His6-MBP-mGFP-YBX1-Δ (1-127) (plasmid) | This study | | To express YBX1-Δ (1-127) in insect cells, plasmid maintained in Schekman lab |
| Recombinant DNA reagent | pX330-Venus (plasmid) | Other | | Gift of Dr. Robert Tjian lab |
| Recombinant DNA reagent | PLKO.1-Puro (plasmid) | Addgene | RRID:Addgene_10878 | Pol III based shRNA backbone |
| Antibody | anti-YBX1 (Rabbit polyclonal) | Cell Signaling Technology | RRID:AB_1950384 | WB (1:1000) |
| Antibody | anti-YBX1 (Rabbit polyclonal) | Abcam | RRID:AB_2219278 | WB (1:1000) IF (1:100) |
| Antibody | anti-CD9 (Rabbit monoclonal) | Cell Signaling Technology | RRID:AB_2798139 | WB (1:3000) |
| Antibody | anti-ALIX (Mouse monoclonal) | Santa Cruz Biotechnology | RRID:AB_673819 | WB (1:1000) |
| Antibody | anti-flotillin-2 (Mouse monoclonal) | BD Biosciences | RRID:AB_397766 | WB (1:1000) |
| Antibody | Anti-CD63 (Mouse monoclonal) | Thermo Fisher Scientific | RRID:AB_2572564 | WB (1:1000) IF (1:100) |
| Antibody | Anti-Actin (Mouse monoclonal) | Abcam | RRID:AB_449644 | WB (1:3000) |
| Antibody | Anti-DDX6 (Rabbit polyclonal) | Bethyl | RRID:AB_2277216 | WB (1:1000) IF (1:100) |
| Antibody | anti-EDC4 (Mouse monoclonal) | Santa Cruz Biotechnology | RRID:AB_10988077 | WB (1:1000) IF (1:50) |
| Antibody | anti-G3BP1 (Mouse monoclonal) | Santa Cruz Biotechnology | RRID:AB_1123055 | WB (1:1000) |
| Antibody | anti-GFP (Mouse monoclonal) | Santa Cruz Biotechnology | RRID:AB_627,695 | WB (1:1000) IF (1:50) |
| Antibody | anti-GFP (Rabbit polyclonal) | Torrey Pines Biolabs | RRID:AB_10013661 | WB (1:1000) IF (1:100) |
| Antibody | anti-GM130 (Mouse monoclonal) | BD Biosciences | RRID:AB_398142 | WB (1:1000) |
| Antibody | anti-Flag (Mouse monoclonal) | Sigma-Aldrich | RRID:AB_439698 | WB (1:3000) |

| Reagent type (species) or resource | Designation | Source or reference | Identifiers | Additional information |
|---|---|---|---|---|
| Antibody | anti-hDcp1a (Mouse monoclonal) | Santa Cruz Biotechnology | RRID:AB_2090408 | IF (1:50) |
| Antibody | anti-LSM14A (Rabbit polyclonal) | Proteintech | RRID:AB_10644339 | WB (1:1000) IF (1:100) |
| Antibody | anti-4E-T (Mouse monoclonal) | Santa Cruz Biotechnology | sc-393788 | WB (1:1000) IF (1:50) |
| Software, algorithm | Fiji | NIH | RRID:SCR_002285 | https://fiji.sc/ |
| Software, algorithm | GraphPad Prism | Other | RRID:SCR_002798 | https://www.graphpad.com |
| Software, algorithm | IUPred | PMID:15955779 | RRID:SCR_014632 | http://iupred.enzim.hu/ |
| Software, algorithm | Sigmaplot 12.5 | Other | RRID:SCR_003210 | http://www.sigmaplot.com/products/sigmaplot/ |
| Software, algorithm | NCPR | Other | | http://www.bioinformatics.nl/cgi-bin/emboss/charge |

## Cell lines and cell culture

Human HEK293T cells and human osteosarcoma cell lines U2OS were obtained from the UC-Berkeley Cell Culture Facility and were confirmed by short tandem repeat profiling (STR) and tested negative for mycoplasma contamination. Cells were grown in monolayer cultures at 37 °C in 5 % $CO_2$ and maintained in Dulbecco's modified Eagle's medium (DMEM) supplemented with 10 % fetal bovine serum (FBS, Thermo Fisher Scientific, Waltham, MA). For EV production, we seeded cells at ~30 % confluency in exosome-depleted medium that was obtained either by ultracentrifugation of DMEM plus 10 % FBS or DMEM supplemented with 10 % exosome-depleted FBS (System Biosciences, Palo Alto, CA) in 150 mm CellBIND tissue culture dishes (Corning, Corning NY). EVs were collected when cells reached approximately 80 % confluency (~48 hr). For characterization of miRNA sorting into exosomes, cells grown to ~70 % confluency in DMEM with 10 % FBS medium and shifted into exosome-depleted medium. EVs were collected after 24 hr.

## Extracellular vesicle purification

Conditioned medium (about 420 ml) was harvested from HEK293T or U2OS cultured cells at 80 % confluency. All the following manipulations were performed at 4 °C. Cells and large debris were removed by centrifugation at 1500xg for 20 min in a Sorvall R6+ centrifuge (Thermo Fisher Scientific) followed by 10,000xg for 20 min in 500 ml vessels using a fixed angle FIBERlite F14−6 × 500 y rotor (Thermo Fisher Scientific). The supernatant fraction was then centrifuged onto a 60 % sucrose cushion in buffer A (10 mM HEPES pH 7.4, 0.85% w/v NaCl) at ~100,000 xg (28,000 rpm) for 1.5 hr using SW 32 Ti swinging-bucket rotors. The interface on the sucrose cushion was collected and pooled from three tubes and applied onto a 60 % sucrose cushion for an additional centrifugation at ~120,000 xg (31,500 rpm) in a SW 41 Ti swinging-bucket rotor for 16 hr. The sucrose concentration of the collection from the first sucrose cushion interface was measured by refractometry and was adjusted to a concentration <20%. Higher concentrations of sucrose impede sedimentation because EVs equilibrate at a buoyant density above that level. For purification, EV subpopulations that resolve at distinct buoyant densities in a linear gradient were collected and mixed with 60 % sucrose to a final volume of 4 ml (sucrose final concentration is ~48%). Layers of 1.5 ml of 25%, 20%, 15%, 10%, and 5 % iodixanol (Optiprep) solution in buffer A were sequentially overlaid and samples were centrifugated at ~150,000xg (36,500 rpm) for 16 hr in a SW 41 Ti rotor. Fractions (400 µl for each) from top to bottom were collected and mixed with SDS sample buffer for immunoblot analysis. In some cases, such as in the immunoblot of YBX1, the floated fractions corresponding to the high density from 1.13 to 1.15 g/ml were pooled and concentrated by centrifugation to improve detection by immunoblot.

For EV purification in bulk (without discriminating among EV sub-populations), the first cushion-sedimented vesicles above were collected and mixed with 60 % sucrose to a final volume of 8 ml.

At this point, it was important to keep the sucrose concentration >50 %. Aliquots (3 ml) of 40%, (1.5 ml)10 % sucrose buffer were sequentially overlaid and the tubes were centrifuged at ~150, 000xg (36,500 rpm) for 16 hr in a SW 41 Ti swinging-bucket rotor. The 10/40 % interface was collected and used either directly for RNA extraction by a mirVana miRNA isolation kit (Thermo Fischer Scientific) or washed with PBS and concentrated by centrifugation at ~120, 000xg in a SW 55 Ti rotor for 70 min. Samples were then prepared for immunoblot analysis.

For proteinase K protection assays, the supernatant fraction from 10,000xg of conditioned medium was centrifuged at 100,000xg (28,000 rpm) for 1.5 hr using SW 32 Ti rotors. Pellet fractions resuspended in PBS were pooled and centrifuged at ~150, 000xg (36,500 rpm) for 70 min in a SW 55 Ti rotor. The pellet was resuspended in PBS and split into four equal aliquots. One sample was left untreated, another sample was treated with 0.5 % Triton X-100, the third sample was treated with 5 μg/ml proteinase K on ice for 20 min, and the last one was mixed with 0.5 % Triton X-100 prior to proteinase K treatment. Proteinase K was inactivated with 5 mM phenylmethane sulfonyl fluoride (PMSF) on ice for 5 min and samples were then mixed with SDS sample loading buffer for immunoblot analysis.

## Nanoparticle tracking analysis

Extracellular vesicles purified by buoyant density centrifugation were diluted 1:100 with PBS filtered with a 0.02 μm filter (What GmbH, Dassel, Germany). The liquid was drawn into a 1 ml syringe and inserted into a Nanosight LM10 instrument equipped with a 405 nM laser (Malvern, UK). Particles were tracked for 60 s at a constant flow rate of 50 using Nanosight nanoparticle tracking analysis software (Nanosight NTA 3.1 software, Malvern Instruments). Each sample was analyzed five times and the counts were averaged.

## Negative staining and visualization of exosomes by electron microscopy

Formvar/Carbon Coated - Copper 300 mesh grids (Electron Microscopy Sciences, Hatfield, PA) were glow discharged for 10 s. An aliquot of exosomes (5 μl) isolated from the 10/40 % interface of the sucrose flotation gradient was spread onto a freshly glow- discharged grid. The grid was quickly washed on three droplets of water to dilute the sucrose and then stained with 1 % uranyl acetate for 2 min. Excess staining solution was removed with Whatman Grade one filter paper. Post drying, grids were imaged at 120 kV using a Tecnai-12 Transmission Electron Microscope (FEI, Hillsboro, OR) in the Electron Microscopy Laboratory at UC Berkeley.

## Quantitative real-time PCR

RNA was extracted using either Direct-zol RNA Miniprep kits (Zymo Research) or a mirVana miRNA isolation kit (Thermo Fisher Scientific) according to the manufacturer's instructions. MiRNAs sequence are as follows: miR-223, UGUCAGUUUGUCAAAUACCCCA; miR-190, UGAUAUGUUUGAUAUAUUAGGU; miR-144, UACAGUAUAGAUGAUGUUACU. Taqman miRNA assays, purchased from Life Technologies (assay number: has-mir-223–3 p:000526, has-mir-190–5 p:000489 and has-mir-144–3 p:002676), were performed to quantify miRNAs. We used total RNA from cells or exosomes for normalization as there is no well-accepted control transcript for exosomes. Total RNA from cells was quantified by nanodrop and total RNA from exosomes was quantified using an RNA bioanalyzer (Agilent). Typically, 10 ng total RNA from cells and 2 ng total RNA from exosomes was reverse transcribed. Taqman qPCR master mix with no amperase UNG was used for real-time PCR and reactions were performed on an ABI-7900 real-time PCR system (Life Technologies). For all RT-PCR reactions, the results are presented as mean cycle threshold (Ct) values of three independent technical replicates. Samples with a Ct value greater than 40 were regarded as negative.

## Measurement of miR-223 secretion

An equal number of cells for indicated cell lines were seeded at 50 % confluency for one day and changed into exosome-depleted media on the second day for the next 24 hr. During this time, 200 μl medium was collected at 2, 6, 12 and 24 hr time points and centrifuged at 1500 xg for 15 min to

remove cellular debris. RNA was extracted using the Direct-zol RNA Miniprep kits (Zymo Research) and analyzed by Taqman miRNA qPCR assay.

## Immunoblots

After washing cells with cold PBS, total cell extracts were isolated in RIPA buffer (50 mM Tris-HCL PH 7.5, 150 mM NaCl, 2 mM EDTA, 1 % Triton X-100) containing a protease inhibitor cocktail (1 mM 4-aminobenzamidine dihydrochloride, 1 mg/ml antipain dihydrochloride, 1 mg/ml aprotinin, 1 mg/ml leupeptin, 1 mg/ml chymostatin, 1 mM phenymethylsulfonly fluoride, 50 mM N-tosyl-L-phenylalanine chloromethyl ketone and 1 mg/ml pepstatin). Typically, around 30–50 µl of RIPA buffer with inhibitor was added per $1 \times 10^6$ cells. Protein was quantified using a BCA Protein Assay Kit (Thermo Fisher Scientific) and appropriate amounts of cell lysate were mixed with SDS sample loading buffer. Samples were heated at 95 °C for 10 min and separated on 4–20% acrylamide Tris-glycine gradient gels (Life Technologies). Proteins were transferred to PVDF membranes (EMD Millipore, Darmstadt, Germany), blocked with 5 % bovine serum albumin in TBST, and incubated for either 2 hr at room temperature or overnight at 4 °C with primary antibodies. Blots were then washed with TBST, incubated with anti-rabbit or anti-mouse secondary antibodies (GE Healthcare Life Sciences, Pittsbugh, PA) and detected with ECL-2 reagent (Thermo Fisher Scientific). Primary antibodies used in this study were as follows: anti-YBX1 (Cell Signaling Technology, Danvers, MA, #4202); anti-YBX1 (Abcam, Cambridge, MA, ab12148); anti-CD9 (Cell Signaling Technology, Danvers, MA, #13,174 S); anti-ALIX (Santa Cruz Biotechnology, CA, Sc-53540); anti-flotillin-2 (BD Biosciences, San Jose, CA, #610383); anti-CD63 (Abcam, Cambridge, MA, ab134045); anti-CD63 (Fisher Scientific, BDB556019, #H5C6); anti-Actin (Abcam, Cambridge, MA, ab8224); anti-DDX6 (Bethyl Laboratories, Inc, A300-461A); anti-EDC4 (Santa Cruz Biotechnology, CA, Sc-376382); anti-G3BP1 (Santa Cruz Biotechnology, CA, Sc-81940); anti-GFP (Torrey Pines Biolabs, Inc, Houston, TX, TP401); anti-GFP (Santa Cruz Biotechnology, CA, Sc-9996); anti-GM130 (BD Biosciences, 610823); anti-Flag (Sigma, St. Louis, MO, F9291).

## Immunoprecipitation of YBX1-miRNA complexes

Immunoprecipitation of YBX1-miRNA complexes was performed as previously described (*Temoche-Diaz et al., 2019*). Briefly, about $4 \times 10^7$ HEK293T cells expressing YFP-tagged YBX1 were harvested. Cells were homogenized in 2 volumes of HB buffer (20 mM HEPES pH7.4, 250 mM sorbitol) with protease inhibitor cocktail as described above and physically disrupted by 13–15 passes through a 22 G needle until ~85% cell lysis was achieved as assessed by trypan blue staining. The homogenate was centrifuged at 1500xg for 20 min to remove unlysed cells and nuclei. The supernatant fraction was used as a source of cytoplasmic YBX1 for immunoprecipitation. GFP-Trap agarose beads (Chromotek) were washed three times in polysome lysis buffer (100 mM KCl, 5 mM MgCl₂, 10 mM HEPES, pH 7.0, 0.5 % Nonidet P-40, 1 mM DTT, 100 U/ ml RNasin RNase inhibitor (Promega, cat. no. N2511), 2 mM vanadyl ribonucleoside complex solution (Sigma-Aldrich (Fluka BioChemika), cat. no. 94742), 1 x protease inhibitor cocktail). We used polysome lysis buffer as described to lyse cells; for washes this buffer was used without RNase and protease inhibitors. The 1,500 xg post-nuclear supernatant was mixed with 5 x polysome lysis buffer to a final 1 x concentration. An aliquot (1/10) of the lysate was set aside as the input and the rest was incubated with 40 µl GFP-Trap beads with rotation for 3 hr at 4 °C. Beads were washed five times with 1 x polysome buffer and divided for protein or RNA analysis. Beads for protein analysis were incubated with SDS sample buffer and heated at 95 °C for 10 min and beads for RNA analysis were mixed with TRI reagent (Zymo Research) followed by RNA extraction using a Direct-zol RNA purification kits (Zymo Research). Proteins and miRNAs were analyzed by immunoblots and Taqman miRNA qPCR, respectively.

## Mass spectrometry analysis of YBX1 immunoprecipitation and purified exosomes

For YBX1 affinity purification, we collected HEK293T cells transfected with either 3xFlag-YBX1 or EGFP-YBX1 which were washed with PBS and resuspended in RIPA buffer containing a protease inhibitor cocktail (as described above). Cells were lysed in RIPA buffer by rotation at 4 °C for 30 min and the cell lysate was clarified by centrifugation at 13,200 rpm for 15 min. The supernatant fraction was

incubated with tag-recognizing antibody beads, which included ANTI-FLAG M2 Affinity Gel (Sigma, A2220) for Flag and GFP-Trap agarose beads (Chromotek) for EGFP, for 3 hr at 4 °C. After incubation, the beads were washed three times using RIPA buffer and another two times using RIPA buffer without 1 % Triton X-100. For Flag beads, bead-bound proteins were eluted in 3xFlag peptide buffer (200 μg/ml 3xFlag peptide, 50 mM Tris-HCl, pH7.4, 150 mM NaCl, 1 x protease inhibitor cocktail, 1 x PMSF) for 30 min at 4 °C. The eluted fraction was heated to 95 °C in Laemmli buffer. For GFP beads, bead-bound proteins were eluted by incubation at 95 °C with Laemmli buffer for 10 min.

Exosomes purified from the sucrose gradient (as *Figure 4G*) were diluted in 12 ml PBS and then centrifuged at ~120,000 xg in an SW41 rotor for 70 min. Sedimented vesicles were heated to 95 °C in Laemmli buffer for 10 min.

For mass spectrometry analysis, heated samples were electrophoresed in a 4–20% acrylamide Tris-Glycine gradient gel (Life Technologies) for about 5 min. The bulk of proteins were stained with Coomassie blue and stained bands were excised from the gel with a fresh razor blade. In order to identify the total proteins from the Coomassie-stained gel section (anything that was not a single gel band), samples were submitted to the Taplin Mass Spectrometry Facility at Harvard Medical School for in-gel tryptic digestion of proteins followed by liquid chromatography and mass spectrometry analysis (LC-MS) according to their standards.

## Immunofluorescence

U2OS cells on 12 mm coverslips (Corning) were washed by PBS once and fixed by 4 % EM-grade paraformadehyde (Electron Microscopy Science, Hatfield, PA) for 15 min at room temperature. Cells were then washed three time with PBS, blocked for 30 min in blocking buffer (5 % FBS in PBS), and permeabilized in blocking buffer with 0.1 % saponin for 20 min. Next, cells were incubated with either 1:50 or 1:100 dilution of primary antibodies for 1 hr at room temperature, washed by PBS three times (10 min/time) and incubated in secondary antibodies with 1: 500 dilution for another 1 hr at room temperature. Cells were extensively washed with PBS another three times (10 min/time) and mounted on slides using Prolong Gold Antifade Reagent with DAPI (Thermo Fisher). Both primary and secondary antibodies were diluted in blocking buffer with 0.1 % saponin. All the incubations were done in a humid light-tight box to prevent drying and fluorochrome fading. Primary antibody uses in the immunofluorescence studies were as follows: anti-YBX1 (Abcam, ab12148); anti-CD63 (Fisher Scientific, BDB556019, #H5C6); anti-DDX6 (Bethyl Laboratories, Inc, A300-461A); anti-EDC4 (Santa Cruz Biotechnology, Sc-376382); anti-Dcp1a (Santa Cruz Biotechnology, Sc-100706); anti-GFP (Torrey Pines Biolabs, TP401); anti-GFP (Santa Cruz Biotechnology, Sc-9996). Images were acquired with a Zeiss LSM710 confocal microscope equipped with an mCherry/GFP/DAPI filter set and 63 X or 100 × 1.4 NA objectives, and were analyzed with the Fiji software (http://fiji.sc/Fiji). In order to better display the results, we adjusted the 'scaling' of the YFP intensity among the panels / mutants in one figure.

## CRISPR/Cas9 genome editing

A pX330-based plasmid expressing venus fluorescent protein (*Shurtleff et al., 2016*) was used to clone the gRNAs targeting YBX1. Two CRISPR guide RNAs targeting the first exon of the YBX1 open reading frame were designed following the CRISPR design website (http://crispor.tefor.net/crispor.py). gRNAs targeting the following sequences within YBX1: YBX1-gRNA1, agcgccgccgacaccaagcc, YBX1-gRNA2, atcggcggcgcctgccggcg; Oligonucleotides encoding gRNAs were annealed and cloned into pX330-Venus as described (Zhang Feng lab's protocol). HEK293T cells at a low passage number were transfected using Lipofectamine 2000 (Invitrogen) for 48 hr, trypsin-treated and sorted for single, venus positive cells in 96 well plates using a BD influx cell sorter. Wells containing single clones (72 clones, 1–24 clones for YBX1-gRNA1, 25–48 clones for YBX1-gRNA2, 49–72 clones for YBX1-gRNA1 and gRNA2) were allowed to expand and YBX1 knockout candidates were confirmed by immunoblot. The YBX1 positive knockouts by immunoblot were clones 9, 31, 41, 54, 57, 58, 66, 67, 68, and 72. Clones 9 and 41 were further verified by Sanger sequencing after Topo TA cloning the PCR products of the region around the gRNA recognition site. U2OS YBX1 knockout cells were generously provided by Dr. Pavel Ivanov (*Lyons et al., 2016*).

## Constructs, protein expression, and purification

Plasmid information is listed in the key resources table. Maltose-binding protein hybrid genes were expressed and the fusion proteins were isolated from baculovirus-infected SF9 insect cells (*Lemaitre et al., 2019*). Insect cell culture (1 L) was harvested 48 hr after viral infection and collected by centrifugation for 15 min at 2000 rpm. The pellet fractions were resuspended in 35 ml lysis buffer (50 mM Tris-HCl 7.4, 1 M KCl, 5 % glycerol, 5 mM MgCl2, 0.5 μl/ml Benzonase nuclease (sigma, 70746–3), 1 mM DTT, 1 mM PMSF and 1 x protease inhibitor cocktail). Cells were lysed by sonication and the crude lysate was clarified by centrifugation for 60 min at 20, 000 rpm at 4 °C. After centrifugation, the supernatant fraction was incubated with 4 ml amylose resin (New England Biolabs, E8021L) for 1 hr. Amylose resin samples were transferred to columns and protein-bound beads were washed with lysis buffer until no protein was eluted as monitored by the Bio-Rad protein assay (Bio-Rad, Catalog #5000006). Maltose-binding protein fusions were eluted with 12 ml elution buffer (50 mM Tris-HCl 7.4, 500 mM KCl, 5 % glycerol, 50 mM maltose) and concentrated using an Amicon Ultra Centrifugal Filter Unit (50 kDa, 4 ml) (Fisher Scientific, EMD Millipore). The protein was further purified by gel filtration chromatography (Superdex-200, GE Healthcare) with columns equilibrated in storage buffer (50 mM Tris-HCl 7.4, 500 mM KCl, 5 % glycerol, 1 mM DTT). Peak fractions corresponding to the appropriate fusion protein were pooled, concentrated and distributed in 5 μl aliquots in PCR tubes, flash-frozen in liquid nitrogen and stored at –80 °C. Protein concentration was determined by measuring absorbance at 280 nm using a NanoDrop ND-1000 spectrophotometer (Thermo Scientific).

In most cases, the above method worked well. However, the 6xHis-MBP-mGFP-YBX1-WT protein fraction contained nucleic acid, which influenced results of the phase separation experiment. For this reason, the following modified protocol was used to purify 6xHis-MBP-mGFP-YBX1-WT protein. Pellet fractions were re-suspended in 35 ml lysis buffer (50 mM Tris-HCl 8.5, 2 M KCl, 5 % glycerol, 10 mM MgCl$_2$, 50 μl Benzonase nuclease, 1 mM DTT, 1 mM PMSF and 1 x protease inhibitor cocktail) and cells were lysed by sonication followed by centrifugation for 60 min at 20,000 rpm at 4 °C. The supernatant fraction was incubated with 4 ml amylose resin for 1 hr. The amylose resins were transferred to columns and protein-bound beads were washed with lysis buffer until no protein, as detected by a Bio-Rad protein assay. Bound proteins were eluted with the elution buffer (50 mM Tris-HCl 8.5, 2 M KCl, 5 % glycerol, 10 mM MgCl$_2$, 50 mM maltose) and concentrated into 500 μl using an Amicon Ultra Centrifugal Filter Unit. Benzonase nuclease (50 μl) was added to digest nucleic acid at 4 °C overnight. The sample was diluted into 5 ml using the elution buffer (50 mM Tris-HCl 8.5, 2 M KCl, 5 % glycerol, 10 mM MgCl$_2$, 50 mM maltose) and passed through 2 ml HisPur Ni-NTA resin (ThermoFisher Scientific, catalog number, 88222). The Ni-NTA column was washed with lysis buffer and the protein was desorbed with 6 ml elution buffer (50 mM Tris-HCl 8.5, 2 M KCl, 5 % glycerol, 250 mM Imidazole) and concentrated into 1 ml using Amicon Ultra Centrifugal Filter Unit. The protein was further purified by gel filtration (Superdex-200, GE Healthcare) in a column equilibrated with storage buffer (50 mM Tris-HCl 7.4, 500 mM KCl, 5 % glycerol, 1 mM DTT). Peak fractions corresponding to the desired protein were pooled, concentrated and aliquoted in PCR tubes, flash-frozen in liquid nitrogen and stored at –80 °C.

## *In vitro* phase separation assays

For droplet formation without crowding agents (*Figure 2A*, up panel, *Figure 2B, C , and D*), proteins were diluted to various concentrations in the buffer containing a final concentration of 25 mM Tris-HCl, pH 7.4, 75 mM KCl at room temperature. For droplet formation in the presence of crowding agents (all the other *in vitro* phase separation experiments), proteins were diluted from a stock solution into buffer containing a final concentration of 5 % dextran, 25 mM Tris-HCl, pH 7.4, 150 mM KCl at room temperature. Proteins were added as the last component to induce uniform phase separation. To observe the propensity of RNA to partition into the condensates, we resuspended RNA in RNase-free water at indicated concentrations. The droplet formation of purified mGFP-YBX1 and Cy5 labeled miRNAs (100 nM miRNAs together with 10 ng/μl total RNA, which is about 100 nM) was induced in LLPS buffer (5 % dextran, 25 mM Tris-HCl, pH 7.4, 150 mM KCl, 1 mM MgCl$_2$ (to stabilizes the RNA secondary structure)). The samples were mixed in a microtube and applied to a coverslip-bottom in 35 mm dishes (MatTek P35G-1.5–14 C). After all the droplets had settled to the bottom, images were taken using an ECLIPSE TE2000 microscope (Nikon) with a 100 x oil-immersion objective.

## Image analysis to determine the partition coefficients and relative condensed protein or miRNAs in *in vitro* droplets

Fluorescence microscopy images were acquired using an ECLIPSE TE2000 microscope with a 100 x oil-immersion objective. Quantification and statistical analysis were performed as previously reported (*Wang et al., 2018*). Images were analyzed using Fiji software. All the results are plotted as the mean ± the standard deviation (SD). A mask of the droplets was defined by threshold of the images and removing spurious noise detection with a median filter window radius equal to two pixels. For the background correction, an image was acquired with the shutter closed and its average intensity was removed from each pixel contribution. The user parameter was initially set to the value three and was adjusted according to the mask of droplets. The partition coefficients (*PCs*) were defined as the average intensity within the dense phase ($I_{DP}$) divided by average intensity in the light phase ($I_{LP}$) after background subtraction. $I_{DP}$ is the average of mean intensities inside the droplets, $I_{LP}$ is the mean intensity of the regions outside the droplets. Four images (1024 × 1024) per condition were analyzed. The relative amount of condensed (*RC*) protein or miRNA was calculated from the equation: RC = $I_{in}$ / $I_{in}$+ $I_{out}$, where $I_{in}$ is the integrated intensity inside the droplets and $I_{out}$ is the total intensity of the region outside the droplet mask. Where no droplet appeared, the value was set to 0. Three images (2048 x 2048) for per condition were analyzed.

## Fluorescence recovery after photobleaching (FRAP)

In vitro droplets were formed by diluting stock protein to a final concentration of 12 µM in 25 mM Tris-HCl, pH 7.4, 150 mM KCl buffer at room temperature. FRAP was performed on an inverted laser scanning confocal microscope (Zeiss, LSM 710 AxioObserver) with 34-channel spectral detection. Images were acquired with a 63 x oil-immersion objective and a 488 nm laser line was used for detection of GFP fluorescence. A circular region of ~1 µm in diameter was chosen in a region away from the droplet boundary and bleached with 13 iterations at ~60 % of maximum laser power at 488 nm. The recovery was recorded at a rate of 30 ms/frame, 40 frames in total. For imaging cells, FRAP was performed using an inverted laser scanning confocal microscope (Zeiss, LSM 880 AxioImager) equipped with a full incubation chamber maintained at 37 °C and supplied with 5 % $CO_2$. The point region was bleached with 10 iterations of 100 % of maximum laser power of a 514 nm laser. The recovery was recorded at the rate of 2 s/interval, 120 cycles in total for YBX1-WT and 1 s/interval, 60 cycles in total for YBX1-F85A. For each sample, a minimum of three independent FRAP experiments were performed. Pictures were analyzed in Fiji software and FRAP recovery curves were calculated as previously described (*Webster et al., 2015*). To account for background and photo-bleaching effects during acquisition, we used the mean intensity values from the bleach region (BL), the background region (BG) and the reference signal region (REF) to calculate the corrected BL (BL_corr) for each acquisition frame using the equation:

BL_corr2 (t) = BL_corr1 (t) / REF_corr1 (t) = [BL(t)-BG(t)] / [REF(t)-BG(t)].

BL_corr2 (t) was further normalized to the mean pre-bleach intensities, which were used to estimate 100 % fluorescence intensity:

BL_corr3 (t) = BL_corr2 (t) / BL_corr2 (pre-bleach).

Finally, an exponential recovery-like-shape curve was generated by plotting the normalized fluorescent intensity value to times.

## Live-cell imaging under hexanediol treatments

Live cell imaging was performed on an LSM880 microscope with the incubation chamber maintained at 37 °C and 5 % $CO_2$. U2OS cells expressing YFP-YBX1 were grown on a coverslip-bottom in 35 mm dishes (MatTek P35G-1.5–14 C) until approximately 70 % confluency and then imaged using the 514 nm laser. The stock solutions of 1,6-hexanediol (Sigma-Aldrich, 240117) and 2,5-hexanediol (Sigma-Aldrich, H11904) with different m/v concentrations (20%, 10%, 4%) in phenol-red free medium were freshly prepared. Right before imaging, the normal cell culture medium was changed into 1 ml of phenol-red-free medium in the 35 mm dish. After starting the image acquisition, we added 1 ml of pre-warmed hexanediol stock solution (20%, 10%, 4%) to the 35 mm dish without pausing imaging to adjust to the final concentrations of 10%, 5%, or 2 %. We

treated the time of hexanediol addition as the time '0' when quantification of surviving puncta in *Figure 1E*.

## Acknowledgements

We thank Dr. Anthony A Hyman for sharing the plasmids; thank Jie Wang for advice and sharing the scripts for statistical analysis of partition coefficient and the amount condensed protein/miRNAs; thank Criss Hartzell, Arup Indra, Matthew J Shurtleff, and Morayma M Temoche-Diaz for suggestions and for reading and editing the manuscript. We also thank the staff at the UC Berkeley shared facilities, the cell culture facility (Alison Killilea), Biological imaging facility (Steven Ruzin), the DNA sequencing facility, the Flow Cytometry Facility, the Electron Microscopy Laboratory and QB3-Berkeley (The California Institute for Quantitative Biosciences at UC Berkeley). XML and LM are supported as Associates of the HHMI. RS is an Investigator of the HHMI and a Senior fellow of the UC Berkeley Miller Institute of Science.

## Additional information

### Competing interests

Randy Schekman: Reviewing editor, *eLife*. The other authors declare that no competing interests exist.

### Funding

| Funder | Grant reference number | Author |
| --- | --- | --- |
| Howard Hughes Medical Institute | Investigator | Randy Schekman |

The funders had no role in study design, data collection and interpretation, or the decision to submit the work for publication.

### Author contributions

Xiao-Man Liu, Conceptualization, Investigation, Methodology, Writing - original draft, Writing - review and editing; Liang Ma, Contribution of unpublished essential data and reagents; Randy Schekman, Conceptualization, Funding acquisition, Investigation, Supervision, Writing - review and editing

### Author ORCIDs

Xiao-Man Liu http://orcid.org/0000-0001-9968-3988
Randy Schekman http://orcid.org/0000-0001-8615-6409

### Decision letter and Author response

Decision letter https://doi.org/10.7554/eLife.71982.sa1
Author response https://doi.org/10.7554/eLife.71982.sa2

## Additional files

### Supplementary files

• Transparent reporting form

### Data availability

All data generated or analysed during this study are included in the manuscript and supporting files. Source data files have been provided for Figure 3-figure supplement 2D, Figure 4C, Figure 4D, Figure 4E, Figure 4F, Figure 4I, Figure 4K, Figure 5C, Figure 7G, Figure 7H, Figure 7I, and Figure 7-figure supplement 3B; Numerical data that are represented as graphs for Figure 5 and Figure 6; Table 1, Table 2, Table 3, and Table 4 as source data corresponding to Figure 7F, Figure 7J, and Figure 7-figure supplement 3.

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
