## [Editor Report]

This paper represents a significant step forward in our understanding how proteins and RNAs are selectively loaded into exosomes, and describes an unexpected cellular use of biomolecular phase separation. This paper will be of interest to molecular cell biologists who study extracellular vesicle biology and liquid-liquid phase separation.

---

## [Decision Letter]

**Decision letter after peer review:**

Thank you for submitting your article "Selective sorting of microRNAs into exosomes by phase-separated YBX1 condensates" for consideration by *eLife*. Your article has been favorably reviewed by 3 peer reviewers, and the evaluation has been overseen by Suzanne Pfeffer as the Senior Editor. The following individual involved in review of your submission has agreed to reveal their identity: Yan G. Zhao (Reviewer #1).

Essential revisions:

All three reviewers were excited about your story and have made some suggestions listed below. The only essential revision is requested to bolster the conclusion, "Condensation of YBX1 into P-bodies is required for sorting miRNAs into exosomes". Please investigate whether disruption of P body formation or P-body and YBX1 interaction affects YBX droplet formation and/or miRNA122 partitioning.

We include the full reviews below so that you can respond to the comments experimentally and/or textually, as you see appropriate and reasonable. We hope you find the comments constructive.

*Reviewer #1 (Recommendations for the authors):*

1. The authors showed that YBX1 condensates into P bodies and 18.4% of the P body components are identified from purified exosomes. As shown Figures 4B, almost all the YBX1 signal are localized inside the enlarged endosomes induced by overexpressing constitutively active form of Rab5, and YBX1 also largely colocalizes EDC4 labeled P-bodies in Figure 7C. What is the localization of different P body components associated with exosomes or not at the same condition? It will be nice to show the segregation of YBX1 droplet from P-bodies before enclosed by endosomes.

2. For the last part of the manuscript, the authors claim that "Condensation of YBX1 into P-bodies is required for sorting miRNAs into exosomes". However, the authors only provide evidence to show the localization and interaction of YBX1 with P-bodies and P-body components in EVs, which fails to demonstrate the requirement of this process in miRNA sorting. The author should investigate whether disruption of P body formation or P-body and YBX1 interaction affects YBX droplet formation and/or miRNA122 partition to support the conclusion.

3. YBX1 has been shown to localize to stress granules. Are stress granule components also identified in exosome proteomics? Is there any relationship between stress granules and exosome sorting? Please discuss.

*Reviewer #2 (Recommendations for the authors):*

I would like to raise several points to be addressed

– Figure 3C, 4A and similar: sorry if I overlooked this, but please describe in the figure legends or methods section how your images were processed. Is e.g. the 'scaling' of the YFP intensity identical for all panels / mutants in one figure, so the expression levels can be directly compared ? Have you checked whether the mutant proteins express to a similar level like the wild-type protein? Protein levels (can) have a great influence on the phase separation behaviour of a protein, so this information would be important for the reader.

– line 216: please show the gel shift assay, for miR-233 but also for at least one other control miRNA (best would be both miR144 and miR190)

– Line 236/237 and 276/277: the authors state that incorporation of YBX1 into ILVs, phase separation and RNA binding of YBX1 are required. While I fully agree on the LLPS part, I am not sure if the statement on RNA binding can be made as is. Figure 3-supp3A shows that the F85A mutant relocates to the nucleus / nucleolus and is depleted from the cytoplasm, which the authors suggest might be a consequence of the deficiency in RNA binding (which could well be). Still, this means that less protein is available the cytoplasm, and the lack of loading might not be due to the deficiency in RNA binding, but simply to cytoplasmic protein availability. One way to test this could be creating a cytoplasm-localised F85A mutant, and testing whether this mutant is recruited to exosomes. Otherwise, please modify the text accordingly.

– Figure 6: it is striking that addition of total RNA (6D) enhances the selectivity / affinity for miR-233 compared to the isolated miRNAs (6-supp2). It would be great if the authors could (a) provide the sequences of the miRNAs and (b) speculate about possible reasons. As an outlook, but not necessarily required for this paper: are certain motifs in miR-233 essential for recruitment (see e.g. recent work by Iserman / Gladtfelter MolCell 2020), and can they distinguish whether the miRNA is recruited to condensates as a single-stranded RNA, or bound to a target mRNA in 6D ? This is also highly interesting in light of the YBX1-DDX6 interaction, since DDX6 is involved in miRNA-mediated repression.

*Reviewer #3 (Recommendations for the authors):*

1. Not essential: A very powerful test of this idea would be to identify experimental conditions to release the F85A mutant (which is retarded in the nucleus) from the nucleus into the cytosol and P bodies and analyze whether miR-223 does indeed fail to get enriched in exosomes. Other mutants that do not affect condensate formation (see Figure 3 supp 2) could be tested as well for miR-233 enrichment.

2. Essential: A link that could be strengthened in this manuscript is the link to processing bodies (P bodies). Previous work identified YBX1 in P bodies and after cells sense stress to stress granules. Indeed, the Y to S as well as the R/K to A mutants do not condense into P bodies, but how does YBX-1 exosome secretion and miR-223 targeting change when P bodies become compromised? Is YBX-1 localization to P bodies a prerequisite for miR-223 sorting?

---

## [Author Response]

Reviewer #1 (Recommendations for the authors):1. The authors showed that YBX1 condensates into P bodies and 18.4% of the P body components are identified from purified exosomes. As shown Figures 4B, almost all the YBX1 signal are localized inside the enlarged endosomes induced by overexpressing constitutively active form of Rab5, and YBX1 also largely colocalizes EDC4 labeled P-bodies in Figure 7C. What is the localization of different P body components associated with exosomes or not at the same condition? It will be nice to show the segregation of YBX1 droplet from P-bodies before enclosed by endosomes.

Under normal condition, localization of different P body components associated with exosomes or not all the same. For example, LSM14A (or EDC4) is not associated with exosomes, DDX6 is associated with exosomes based on the LC-MS/MS results. They both form puncta together with YBX1 (Figure 7A and Figure 7B). In cells overexpressing a constitutively active mutant mCherry-Rab5^Q79L^, YBX1 accumulated inside enlarged endosomes, DDX6 and LSM14A localized either outside of the enlarged endosomes (white arrow) or on the rim of the enlarged endosomes (yellow arrow). These results were shown in “Author response image 1”, which is not included in the revised manuscript.

2. For the last part of the manuscript, the authors claim that "Condensation of YBX1 into P-bodies is required for sorting miRNAs into exosomes". However, the authors only provide evidence to show the localization and interaction of YBX1 with P-bodies and P-body components in EVs, which fails to demonstrate the requirement of this process in miRNA sorting. The author should investigate whether disruption of P body formation or P-body and YBX1 interaction affects YBX droplet formation and/or miRNA122 partition to support the conclusion.

We thank the reviewer for pointing out this important issue. In Figure 7C and Figure 7H, we showed that IDR defective mutants YBX1-CTD-Y to S and YBX1-CTD-RK to G disrupt the association of YBX1 with P-bodies. These IDR defective mutants greatly impaired YBX1 droplet formation (Figure 3D) and miR-223 partition (Figure 6B and Figure 6C).

Treatment of cells with the translation inhibitor CHX (cycloheximide) is well established to trap mRNAs in polysomes and prevent P-body assembly. To test whether P-bodies are involved in exosomal miRNA sorting, we treated U2OS cells with 10 µg/ml CHX. The results are shown in Figure 7 supplement 2 in the revised manuscript. We found that both the P body marker EDC4 and YBX1 were no longer observed in foci following treatment with 10 µg/ml CHX for 2 h (Figure 7 supplement 2A), and miR-223 sorting into the growth medium was reduced significantly (Figure 7 supplement 2B) and by 2-fold into exosomes (Figure 7 supplement 2D) as determined by RT-qPCR.

Three proteins have been reported to be essential for P-body formation in mammalian cells: DDX6, 4E-T, and, to a lesser extent, LSM14A (Ayache et al., 2015; Standart and Weil, 2018). We used shRNA to knock down each protein and found that P-body formation was greatly disrupted in most of the knock down cells (Figure 7 supplement 3A, type A cells). We observed a defect in sorting of miR-223 into isolated exosomes from 4E-T knockdown cells but less so from LSM14A not at all from DDX6 or knockdown cells (Figure 7 supplement 3C). One possible reason is that the knock down for DDX6 and to some extent for LSM14A was incomplete (Figure 7 supplement 3B) and small granules (type A cells) or even big granules (type B cells) were observed in the DDX6 or LSM14A knock downs (Figure 7 supplement 3A). These results are now included in the revised manuscript and described in the Discussion section.

3. YBX1 has been shown to localize to stress granules. Are stress granule components also identified in exosome proteomics? Is there any relationship between stress granules and exosome sorting? Please discuss.

Yes. We detected stress granule components in purified exosomes and added the results in Table 3 and are described in the Discussion section of the revised manuscript. Compared with the published stress granule proteome (Jain et al., 2016), we found that 28.7% (118/411) of stress granule proteins were identified in exosomes. YBX1 enrichment in stress granules has been reported to be linked to its extracellular accumulation during arsenite treatment (Guarino et al., 2018), suggesting a possible role of stress granules in exosome sorting.

Reviewer #2 (Recommendations for the authors):I would like to raise several points to be addressed– Figure 3C, 4A and similar: sorry if I overlooked this, but please describe in the figure legends or methods section how your images were processed. Is e.g. the 'scaling' of the YFP intensity identical for all panels / mutants in one figure, so the expression levels can be directly compared ? Have you checked whether the mutant proteins express to a similar level like the wild-type protein? Protein levels (can) have a great influence on the phase separation behaviour of a protein, so this information would be important for the reader.

In Figure 3C, 4A and similar, cells displayed heterogeneous signals due to the use of transient transfection, but showed similar signal distributions for individual samples. Cells with median level of expression of YFP were chosen for further analysis. We added image processing information in the “Immunofluorescence” section in Methods in the revised manuscript: “Images were acquired with a Zeiss LSM710 confocal microscope equipped with an mCherry/GFP/DAPI filter set and 63X or 100X 1.4 NA objectives, and were analyzed with the Fiji software (http://fiji.sc/Fiji). In order to better display the results, we adjusted the 'scaling' of the YFP intensity among the panels / mutants in one figure.”

The expression levels of each mutant and WT for Figure 3C and Figure 3—figure supplement 2B and 2C are added in the Figure 3 supplement 2F in the revised manuscript. We also added one sentence in the text: “…All the mutants expressed similar or higher levels of the fusion protein compared to wild type (Figure 3 supplement 2F)…” However, the immunoblot results represent the protein expression level of the heterogeneous cell population but not an individual signal cell as shown by images. To further confirm the phase separation mutant phenotype, we isolated single stable cell lines that express similar levels of mutant and wild-type proteins (Figure 4D) and corresponding images are shown in Figure 7C.

– line 216: please show the gel shift assay, for miR-233 but also for at least one other control miRNA (best would be both miR144 and miR190)

The gel shift assay is shown in Author response image 2, but is not included in the revised manuscript. This work will be prepared for a separate publication (Ma, L. and RS, in preparation).

– Line 236/237 and 276/277: the authors state that incorporation of YBX1 into ILVs, phase separation and RNA binding of YBX1 are required. While I fully agree on the LLPS part, I am not sure if the statement on RNA binding can be made as is. Figure 3-supp3A shows that the F85A mutant relocates to the nucleus / nucleolus and is depleted from the cytoplasm, which the authors suggest might be a consequence of the deficiency in RNA binding (which could well be). Still, this means that less protein is available the cytoplasm, and the lack of loading might not be due to the deficiency in RNA binding, but simply to cytoplasmic protein availability. One way to test this could be creating a cytoplasm-localised F85A mutant, and testing whether this mutant is recruited to exosomes. Otherwise, please modify the text accordingly.

We agree with the reviewer’s opinion on the explanation of F85A mutant phenotype. Three nuclear localization signals have been identified in the YBX1 CTD (NLS-1: 149-156aa; NLS-2: 185-194aa; NLS-3: 276-292aa) (Roeyen et al., 2013). The NLS-2 and NLS-3 motifs are sufficient for nuclear targeting. Four related constructs P_NLS_185I, P_NLS_185II, P_NLS_276I, and P_NLS_276II failed to localize to the nucleus (Roeyen et al., 2013). In order to create a cytoplasm-localized F85A mutant, we generated five constructs as shown in author response image 3**.** Although ΔNLS-2, ΔP_NLS_185I, ΔNLS-3 and ΔP_NLS_276II mutants showed partial cytoplasmic localization, none of them resulted in a clean defect. Given this ambiguity, we have not included these results in the revised manuscript and instead have modified our conclusion. We added one sentence “Because the F85A mutant protein mostly localizes to the nucleus, it was used as a control in the following assays.” in line 228-230 in the revised manuscript. We also deleted the words “and RNA binding” in line 248, “both binding to RNA and” in line 290, “RNA-binding CSD and the” in line 310, and “RNA binding and” in line 315 in the revised manuscript.

– Figure 6: it is striking that addition of total RNA (6D) enhances the selectivity / affinity for miR-233 compared to the isolated miRNAs (6-supp2). It would be great if the authors could (a) provide the sequences of the miRNAs and (b) speculate about possible reasons. As an outlook, but not necessarily required for this paper: are certain motifs in miR-233 essential for recruitment (see e.g. recent work by Iserman / Gladtfelter MolCell 2020), and can they distinguish whether the miRNA is recruited to condensates as a single-stranded RNA, or bound to a target mRNA in 6D ? This is also highly interesting in light of the YBX1-DDX6 interaction, since DDX6 is involved in miRNA-mediated repression.

We added the miRNA sequence information in “Quantitative real-time PCR” in the methods section in the revised manuscript: “…MiRNAs sequence are as follows: miR-223, UGUCAGUUUGUCAAAUACCCCA; miR-190, UGAUAUGUUUGAUAUAUUAGGU; miR-144, UACAGUAUAGAUGAUGUACU…”. Our data showed that a low amount of total cellular RNA enhanced and high levels dissolved YBX1 condensates. The condensation of YBX1 peaks at the RNA concentration at which charges between protein and RNA are balanced. In this condition, the YBX1 droplets showed more selectivity for miR-223 over miR-190 or miR-144 due to YBX1-CSD-mediated specific protein RNA interaction.

Our unpublished results show that “UCAGU” motif on the N-terminal of miR-223 is required for its interaction with YBX1 and sorting into exosomes (Ma, L. and RS, in preparation). We speculate that this motif may be essential for its recruitment. In Figure 6 supplement 2B and 2C, there is no cellular total RNA in the phase separation reaction, thus the miRNA may be recruited to condensates as a single-stranded molecule in vitro.

Reviewer #3 (Recommendations for the authors):1. Not essential: A very powerful test of this idea would be to identify experimental conditions to release the F85A mutant (which is retarded in the nucleus) from the nucleus into the cytosol and P bodies and analyze whether miR-223 does indeed fail to get enriched in exosomes. Other mutants that do not affect condensate formation (see Figure 3 supp 2) could be tested as well for miR-233 enrichment.

We thank the reviewer for the suggestion. In author response image 3, we generated five NLS mutants to release the F85A mutant from the nucleus into the cytosol. None of them eliminated nuclear localization. We are sorry that we might not be able to test the idea that reviewer suggested in a short term. We instead modified the text as described above.

2. Essential: A link that could be strengthened in this manuscript is the link to processing bodies (P bodies). Previous work identified YBX1 in P bodies and after cells sense stress to stress granules. Indeed, the Y to S as well as the R/K to A mutants do not condense into P bodies, but how does YBX-1 exosome secretion and miR-223 targeting change when P bodies become compromised? Is YBX-1 localization to P bodies a prerequisite for miR-223 sorting?

We thank the reviewer for pointing out this important issue. We used two methods to dissolve P bodies: CHX treatment (Figure 7 supplement 2) and knock down of key components of the P body (Figure 7 supplement 3) which are now included in the revised manuscript. The results show that disruption of P bodies results in reduced miR-223 secretion as described above. YBX1-CTD-Y to S and YBX1-CTD-RK to G mutants disrupt the localization of YBX1 to P bodies and result in an miR-223 sorting defect, suggesting that YBX1 localization to P bodies is involved in miR-223 sorting.

**References**

Ayache, J., Bénard, M., Ernoult-Lange, M., Minshall, N., Standart, N., Kress, M., and Weil, D. (2015). P-body assembly requires DDX6 repression complexes rather than decay or Ataxin2/2L complexes. Mol. Biol. Cell *26*, 2579–2595.

Guarino, A.M., Troiano, A., Pizzo, E., Bosso, A., Vivo, M., Pinto, G., Amoresano, A., Pollice, A., La Mantia, G., and Calabrò, V. (2018). Oxidative Stress Causes Enhanced Secretion of YB-1 Protein that Restrains Proliferation of Receiving Cells. Genes *9*, 513.

Jain, S., Wheeler, J.R., Walters, R.W., Agrawal, A., Barsic, A., and Parker, R. (2016). ATPase-Modulated Stress Granules Contain a Diverse Proteome and Substructure. Cell *164*, 487–498.

Palazzo, A.F., and Lee, E.S. (2015). Non-coding RNA: what is functional and what is junk? Front. Genet. *6*, 2.

Roeyen, C.R. van, Scurt, F.G., Brandt, S., Kuhl, V.A., Martinkus, S., Djudjaj, S., Raffetseder, U., Royer, H.-D., Stefanidis, I., Dunn, S.E., et al. (2013). Cold shock Y-box protein-1 proteolysis autoregulates its transcriptional activities. Cell Commun. Signal. CCS *11*, 63.

Spannl, S., Tereshchenko, M., Mastromarco, G.J., Ihn, S.J., and Lee, H.O. (2019). Biomolecular condensates in neurodegeneration and cancer. Traffic *20*, 890–911.

Standart, N., and Weil, D. (2018). P-Bodies: Cytosolic Droplets for Coordinated mRNA Storage. Trends Genet. *34*, 612–626.

Woodruff, J.B., Hyman, A.A., and Boke, E. (2018). Organization and Function of Non-dynamic Biomolecular Condensates. Trends Biochem. Sci. *43*, 81–94.